# Antigen-Specific Antibody Design via Direct Energy-based Preference Optimization

**Xiangxin Zhou**[1,2,4,*]  **Dongyu Xue**[4,*]  **Ruizhe Chen**[3,4,*]

**Zaixiang Zheng**[4]  **Liang Wang**[1,2]  **Quanquan Gu**[4,†]

[1]School of Artificial Intelligence, University of Chinese Academy of Sciences
[2]New Laboratory of Pattern Recognition (NLPR),
State Key Laboratory of Multimodal Artificial Intelligence Systems (MAIS),
Institute of Automation, Chinese Academy of Sciences (CASIA)
[3]College of Computer Science and Electronic Engineering, Hunan University
[4]ByteDance Research

## Abstract

Antibody design, a crucial task with significant implications across various disciplines such as therapeutics and biology, presents considerable challenges due to its intricate nature. In this paper, we tackle antigen-specific antibody sequence-structure co-design as an optimization problem towards specific preferences, considering both rationality and functionality. Leveraging a pre-trained conditional diffusion model that jointly models sequences and structures of antibodies with equivariant neural networks, we propose *direct energy-based preference optimization* to guide the generation of antibodies with both rational structures and considerable binding affinities to given antigens. Our method involves fine-tuning the pre-trained diffusion model using a residue-level decomposed energy preference. Additionally, we employ gradient surgery to address conflicts between various types of energy, such as attraction and repulsion. Experiments on RAbD benchmark show that our approach effectively optimizes the energy of generated antibodies and achieves state-of-the-art performance in designing high-quality antibodies with low total energy and high binding affinity simultaneously, demonstrating the superiority of our approach.

## 1 Introduction

Antibodies, vital proteins with an inherent Y-shaped structure in the immune system, are produced in response to an immunological challenge. Their primary function is to discern and neutralize specific pathogens, typically referred to as antigens, with a significant degree of specificity [39]. The specificity mainly comes from the Complementarity Determining Regions (CDRs), which accounts for most binding affinity to specific antigens [24, 15, 49, 2]. Hence, the design of CDRs is a crucial step in developing potent therapeutic antibodies, which plays an important role in drug discovery.

Traditional *in silico* antibody design methods rely on sampling or searching protein sequences over a large search space to optimize the physical and chemical energy, which is inefficient and easily trapped in bad local minima [1, 31, 47]. Recently, deep generative models have been employed to model protein sequences in nature for antibody design [5, 17]. Following the fundamental biological principle

---

*Equal contribution (this work was done during Xiangxin and Ruizhe's internship at ByteDance Research).
†Correspondence to: Quanquan Gu <quanquan.gu@bytedance.com>.

that structure determines function numerous efforts have been focused on antibody sequence-structure co-design [22, 21, 36, 29, 30, 37], which demonstrate superiority over sequence design-based methods.

However, the main evaluation metrics in the aforementioned works are amino acid recovery (AAR) and root mean square deviation (RMSD) between the generated antibody and the real one. This is controversial because AAR is susceptible to manipulation and does not precisely gauge the quality of the generated antibody sequence. Meanwhile, RMSD does not involve side chains, which are vital for antigen-antibody interaction. Besides, it is biologically plausible that a specific antigen can potentially bind with multiple efficacious antibodies [45, 12]. This motivates us to examine the generated structures and sequences of antibodies through the lens of energy, which reflects the rationality of the designed antibodies and their binding affinity to the target antigens. We have noted that nearly all antibody

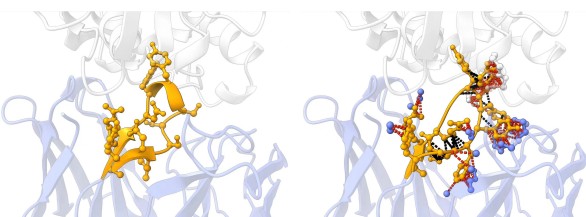

Figure 1: The third CDR in the heavy chain, CDR-H3 (colored in yellow), of real antibody (left) and synthetic antibody (right) designed by MEAN [29] for a given antigen (PDB ID: 4cmh). The rest parts of antibodies except CDR-H3 are colored in blue. The antigens are colored in gray. We use red (resp. black) dotted lines to represent clashes between a CDR-H3 atom and a framework/antigen atom (resp. another CDR-H3 atom). We consider a clash occurs when the overlap of the van der Waals radii of two atoms exceeds 0.6Å.

sequence-structure co-design methods struggle to produce antibodies with low energy. This suggests the presence of irrational structures and inadequate binding affinity in antibodies designed by these methods (see Fig. 1). We attribute this incapability to the insufficient model training caused by a scarcity of high-quality data.

To tackle the above challenges and bridge the gap between *in silico* antibody sequence-structure co-design methods and the intrinsic need for drug discovery, we formulate the antibody design task as an antibody optimization problem with a focus on better rationality and functionality. Inspired by direct preference optimization [DPO, 41] and self-play fine-tuning techniques [10] that achieve huge success in the alignment of large language models (LLMs), we proposed a direct energy-based preference optimization method named ABDPO for antibody optimization. More specifically, we first pre-train a conditional diffusion model on real antigen-antibody datasets, which simultaneously captures sequences and structures of complementarity-determining regions (CDR) in antibodies with equivariant neural networks. We then progressively fine-tune this model using synthetic antibodies generated by the model itself given an antigen with energy-based preference. This preference is defined at a fine-grained residue level, which promotes the effectiveness and efficiency of the optimization process. To fulfill the requirement of various optimization objectives, we decompose the energy into multiple types so that we can incorporate prior knowledge and mitigate the interference between conflicting objectives (e.g., repulsion and attraction energy) to guide the optimization process. Fine-tuning with self-synthesized energy-based antibody preference data represents a revolutionary solution to address the limitation of scarce high-quality real-world data, a significant challenge in this domain. We highlight our main contributions as follows:

- We tackle the antibody sequence-structure co-design problem through the lens of energy from the perspectives of both rationality and functionality.
- We propose direct residue-level energy-based preference optimization to fine-tune diffusion models for designing antibodies with rational structures and high binding affinity to specific antigens.
- We introduce energy decomposition and conflict mitigation techniques to enhance the effectiveness and efficiency of the optimization process.
- Experiments show ABDPO's effectiveness in generating antibodies with energies resembling natural antibodies and generality in optimizing multiple preferences.

## 2 Related Work

**Antibody Design.** The application of deep learning to antibody design can be traced back to at least [35, 43, 3]. In recent years, sequence-structure co-design of antibodies has attracted increasing attention. Jin et al. [22] proposed to simultaneously design sequences and structures of CDRs in an autoregressive way and iteratively refine the designed structures. Jin et al. [21] further utilized the epitope and focused on designing CDR-H3 with a hierarchical message passing equivariant

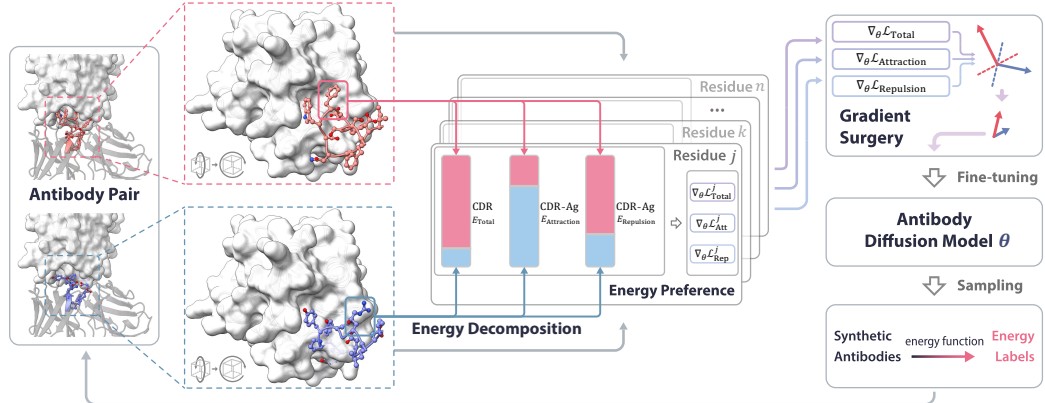

Figure 2: Overview of ABDPO. This process can be summarized as: (a) Generate antibodies with the pre-trained diffusion model; (b) Evaluate the multiple types of residue-level energy and construct preference data; (c) Compute the losses for energy-based preference optimization and mitigate the conflicts between losses of multiple types of energy; (d) Update the diffusion model.

network. Kong et al. [29] incorporated antigens and the light chains of antibodies as conditions and designed CDRs with E(3)-equivariant graph networks via a progressive full-shot scheme. Luo et al. [36] proposed a diffusion model that takes residue types, atom coordinates and side-chain orientations into consideration to generate antigen-specific CDRs. Kong et al. [30] focused on epitope-binding CDR-H3 design and modelled full-atom geometry. Recently, Martinkus et al. [37] proposed AbDiffuser, a novel diffusion model for antibody design, that incorporates more domain knowledge and physics-based constraints and also enables side-chain generation. Besides, Wu and Li [48], Gao et al. [19] and Zheng et al. [52] introduced pre-trained protein language model to antibody design. Distinct from the above works, our method places a stronger emphasis on designing and optimizing antibodies with low energy and high binding affinity.

**Alignment of Generative Models.** Solely maximizing the likelihood of training data does not always lead to a model that satisfies users' preferences. Recently, many efforts have been made on the alignment of the generative models to human preferences. Reinforcement learning has been introduced to learning from human/AI feedback to large language models, such as RLHF [40] and RLAIF [33]. Typically, RLHF consists of three phases: supervised fine-tuning, reward modeling, and RL fine-tuning. Similar ideas have also been introduced to text-to-image generation, such as DDPO [7], DPOK [16] and DiffAC [53]. They view the generative processes of diffusion models as a multi-step Markov Decision Process (MDP) and apply policy gradient for fine-tuning. Rafailov et al. [41] proposed direct preference optimization (DPO) to directly fine-tune language models on preference data, which matches RLHF in performance. Recently, DPO has been introduced to text-to-image generation [46, 6]. Notably, in the aforementioned works, models pre-trained with large-scale datasets have already shown strong performance, in which case alignment further increases users' satisfaction. In contrast, in our work, the model pre-trained with limited real-world antibody data is insufficient in performance. Therefore, preference optimization in our case is primarily used to help the model understand the essence of nature and meet the requirement of antibody design.

## 3 Method

In this section, we present ABDPO, a direct energy-based preference optimization method for designing antibodies with reasonable rationality and functionality (Fig. 2). We first define the antibody generation task and introduce the diffusion model for this task in Sec. 3.1. Then we introduce residue-level preference optimization for fine-tuning the diffusion model and analyze its advantages in effectiveness and efficiency in Sec. 3.2. Finally, in Sec. 3.3, we introduce the energy decomposition and describe how to mitigate the conflicts when optimizing multiple types of energy.

### 3.1 Preliminaries

We focus on designing CDR-H3 of the antibody given antigen structure as CDR-H3 contributes the most to the diversity and specificity of antibodies [49, 2] and the rest part of the antibody including the frameworks and other CDRs. Following Luo et al. [36], each amino acid is represented by its type $s_i \in \{ACDEFGHIKLMNPQRSTVWY\}$, $C_\alpha$ coordinate $\mathbf{x}_i \in \mathbb{R}^3$, and frame orientation

$\mathbf{O}_i \in \text{SO}(3)$ [28], where $i = 1, \ldots, N$ and $N$ is the number of the amino acids in the protein complex. We assume the CDR-H3 to be generated has $m$ amino acids, which can be denoted by $\mathcal{R} = \{(\text{s}_j, \mathbf{x}_j, \mathbf{O}_j) | j = n+1, \ldots, n+m\}$, where $n + 1$ is the index of the first residue in CDR-H3 sequence. The rest part of the antigen-antibody complex can be denoted by $\mathcal{P} = \{(\text{s}_i, \mathbf{x}_i, \mathbf{O}_i) | i \in \{1, \ldots, N\} \setminus \{n+1, \cdots, n+m\}\}$. The antibody generation task can be then formulated as modeling the conditional distribution $P(\mathcal{R}|\mathcal{P})$.

Denoising Diffusion Probabilistic Model [DDPM, 20] have been introduced to antibody generation by Luo et al. [36]. This approach consists of a forward *diffusion* process and a reverse *generative* process. The diffusion process gradually injects noises into data as follows:

$$q(\text{s}_j^t|\text{s}_j^0) = \mathcal{C}\left(\mathbb{1}(\text{s}_j^t)\big|\bar{\alpha}^t \mathbb{1}(\text{s}_j^0) + \bar{\beta}^t \mathbb{1}/K\right),$$

$$q(\mathbf{x}_j^t|\mathbf{x}_j^0) = \mathcal{N}\left(\mathbf{x}_j^t\big|\sqrt{\bar{\alpha}^t}\mathbf{x}_j^0, \bar{\beta}^t \mathbf{I}\right),$$

$$q(\mathbf{O}_j^t|\mathbf{O}_j^0) = \mathcal{IG}_{\text{SO}(3)}\left(\mathbf{O}_j^t\big|\texttt{ScaleRot}\left(\sqrt{\bar{\alpha}_t}\mathbf{O}_j^0\right), \bar{\beta}^t\right),$$

where $(\text{s}_j^0, \mathbf{x}_j^0, \mathbf{O}_j^0)$ are the noisy-free amino acid at time step 0 with index $j$, and $(\text{s}_j^t, \mathbf{x}_j^t, \mathbf{O}_j^t)$ are the noisy amino acid at time step $t$. $\mathbb{1}(\cdot)$ is the one-hot operation. $\{\beta^t\}_{t=1}^T$ is the noise schedule for the diffusion process [20], and we define $\bar{\alpha}^t = \prod_{\tau=1}^{t}(1 - \beta^\tau)$ and $\bar{\beta}^t = 1 - \bar{\alpha}^t$. $K$ is the number of amino acid types. Here, $\mathcal{C}(\cdot)$, $\mathcal{N}(\cdot)$, and $\mathcal{IG}_{\text{SO}(3)}(\cdot)$ are categorical distribution, Gaussian distribution on $\mathbb{R}^3$, and isotropic Gaussian distribution on SO(3) [32] respectively. $\texttt{ScaleRot}$ scales the rotation angle with fixed rotation axis to modify the rotation matrix [18].

Correspondingly, the reverse generative process learns to recover data by iterative denoising. The denoising process $p(\mathcal{R}^{t-1}|\mathcal{R}^t, \mathcal{P})$ from time step $t$ to time step $t-1$ is defined as follows:

$$p(\text{s}_j^{t-1}|\mathcal{R}^t, \mathcal{P}) = \mathcal{C}(\text{s}_j^{t-1}\big|\boldsymbol{f}_{\boldsymbol{\theta}_1}(\mathcal{R}^t, \mathcal{P})[j]), \tag{1}$$

$$p(\mathbf{x}_j^{t-1}|\mathcal{R}^t, \mathcal{P}) = \mathcal{N}(\mathbf{x}_j^{t-1}\big|\boldsymbol{f}_{\boldsymbol{\theta}_2}(\mathcal{R}^t, \mathcal{P})[j], \beta^t \mathbf{I}), \tag{2}$$

$$p(\mathbf{O}_j^{t-1}|\mathcal{R}^t, \mathcal{P}) = \mathcal{IG}_{\text{SO}(3)}(\boldsymbol{f}_{\boldsymbol{\theta}_3}(\mathcal{R}^t, \mathcal{P})[j], \beta^t), \tag{3}$$

where $\mathcal{R}^t = \{\text{s}_j, \mathbf{x}_j, \mathbf{O}_j\}_{j=n+1}^{n+m}$ is the noisy sequence and structure of CDR-H3 at time step $t$, $\boldsymbol{f}_{\boldsymbol{\theta}_1}, \boldsymbol{f}_{\boldsymbol{\theta}_2}, \boldsymbol{f}_{\boldsymbol{\theta}_3}$ are parameterized by denoising neural networks. We utilize SE(3)-equivariant networks [23, 25] as denoising networks because proteins are structures in three-dimensional space, and their properties and characteristics should remain invariant regardless of their observation views. $\boldsymbol{f}(\cdot)[j]$ denotes the output that corresponds to the $j$-th amino acid. The training objective of the reverse generative process is to minimize the Kullback–Leibler (KL) divergence between the variational distribution $p$ and the posterior distribution $q$ as follows:

$$L = \mathbb{E}_{\mathcal{R}^t \sim q}\left[\frac{1}{m}\sum_{j=n+1}^{n+m} \mathbb{D}_{\text{KL}}\Big(q(\mathcal{R}^{t-1}[j]|\mathcal{R}^t, \mathcal{R}^0, \mathcal{P})\big\|p_{\boldsymbol{\theta}}(\mathcal{R}^{t-1}[j]|\mathcal{R}^t, \mathcal{P})\Big)\right]. \tag{4}$$

With some algebra, we can simplify the above objective and derive the reconstruction loss at time step $t$ as follows:

$$L_{\text{s}}^t = \mathbb{E}_{\mathcal{R}^t}\left[\frac{1}{m}\sum_{j=n+1}^{n+m} \mathbb{D}_{\text{KL}}\big(q(\text{s}_j^{t-1}|\text{s}_j^t, \text{s}_j^0)\big\|p(\text{s}_j^{t-1}|\mathcal{R}^t, \mathcal{P})\big)\right], \tag{5}$$

$$L_{\mathbf{x}}^t = \mathbb{E}_{\mathcal{R}^t}\left[\frac{1}{m}\sum_{j=n+1}^{n+m} \big\|\mathbf{x}_j^0 - \boldsymbol{f}_{\boldsymbol{\theta}_2}(\mathcal{R}^t, \mathcal{P})\big\|^2\right], \tag{6}$$

$$L_{\mathbf{O}}^t = \mathbb{E}_{\mathcal{R}^t}\left[\frac{1}{m}\sum_{j=n+1}^{n+m} \big\|(\mathbf{O}_j^0)^\intercal \boldsymbol{f}_{\boldsymbol{\theta}_3}(\mathcal{R}^t, \mathcal{P})[j] - \mathbf{I}\big\|_F^2\right], \tag{7}$$

where $\mathcal{R}^t \sim q(\mathcal{R}^t|\mathcal{R}^0)$ and $\mathcal{R}^0 \sim P(\mathcal{R}|\mathcal{P})$, and $\|\cdot\|_F$ is the matrix Frobenius norm. Note that as Luo et al. [36] mentioned, Eqs. (1) and (3) are an empirical perturbation-denosing process instead of a rigorous one. Thus the terminology *KL-divergence* may not be proper for orientation $\mathbf{O}$. Nevertheless, we can still approximately derive an empirical reconstruction loss for orientation $\mathbf{O}$ as above that works in practice. The overall loss is $L \approx \mathbb{E}_{t \sim \text{U}[1,T]}[L_{\text{s}}^t + L_{\mathbf{x}}^t + L_{\mathbf{O}}^t]$. After optimizing this loss, we can start with the noises from the prior distribution and then apply the reverse process to generate antibodies.

## 3.2 Direct Energy-based Preference Optimization

Only the antibodies with considerable sequence-structure rationality and binding affinity can be used as effective therapeutic candidates. Fortunately, these two properties can be estimated by biophysical energy. Thus, we introduce direct energy-based preference optimization to fine-tune the pre-trained diffusion models for antibody design.

Inspired by RLHF [40], we can fine-tune the pre-trained model to maximize the reward as:

$$\max_{\boldsymbol{\theta}} \mathbb{E}_{\mathcal{R}^0 \sim p_{\boldsymbol{\theta}}}[r(\mathcal{R}^0)] - \beta \mathbb{D}_{\mathrm{KL}}(p_{\boldsymbol{\theta}}(\mathcal{R}^0) \| p_{\mathrm{ref}}(\mathcal{R}^0)),$$

where $p_{\boldsymbol{\theta}}$ (resp. $p_{\mathrm{ref}}$) is the distribution induced by the model being fine-tuned (resp. the fixed pre-trained model), $\beta$ is a hyperparameter that controls the KL divergence regularization, and $r(\cdot)$ is the reward function. The optimal solution to the above objective takes the form:

$$p_{\theta^*}(\mathcal{R}^0) = \frac{1}{Z} p_{\mathrm{ref}}(\mathcal{R}^0) \exp\left(\frac{1}{\beta} r(\mathcal{R}^0)\right).$$

Following Rafailov et al. [41], we turn to the DPO objective as follows:

$$L_{\mathrm{DPO}} = -\mathbb{E}_{\mathcal{R}_1^0, \mathcal{R}_2^0}\left[\log \sigma\left(\beta \mathrm{sgn}(\mathcal{R}_1^0, \mathcal{R}_2^0)\left[\log \frac{p_{\boldsymbol{\theta}}(\mathcal{R}_1^0)}{p_{\mathrm{ref}}(\mathcal{R}_1^0)} - \log \frac{p_{\boldsymbol{\theta}}(\mathcal{R}_2^0)}{p_{\mathrm{ref}}(\mathcal{R}_2^0)}\right]\right)\right],$$

where $\sigma(\cdot)$ is sigmoid and $\mathrm{sgn}(\mathcal{R}_1^0, \mathcal{R}_2^0)$ indicate the preference over $\mathcal{R}_1^0$ and $\mathcal{R}_2^0$. We use "$\succ$" to denote the preference. Specifically, $\mathrm{sgn}(\mathcal{R}_1^0, \mathcal{R}_2^0) = 1$ (resp. $-1$) if $\mathcal{R}_1^0 \succ \mathcal{R}_2^0$ (resp. $\mathcal{R}_2^0 \prec \mathcal{R}_1^0$) in which case we call $\mathcal{R}_1^0$ (resp. $\mathcal{R}_2^0$) the "winning" sample and $\mathcal{R}_2^0$ (resp. $\mathcal{R}_1^0$) the "losing" sample, and $\mathrm{sgn}(\mathcal{R}_1^0, \mathcal{R}_2^0) = 0$ if they tie. $\mathcal{R}_1^0$ and $\mathcal{R}_2^0$ are a pair of data sampled from the Bradley-Terry [BT, 8] model with reward $r(\cdot)$, i.e., $p(\mathcal{R}_1^0 \succ \mathcal{R}_2^0) = \sigma(r(\mathcal{R}_1^0) - r(\mathcal{R}_2^0))$. Please refer to Appendix C for more detailed derivations.

Due to the intractable $p_{\boldsymbol{\theta}}(\mathcal{R}^0)$, following Wallace et al. [46], we introduce latent variables $\mathcal{R}^{1:T}$ and utilize the evidence lower bound optimization (ELBO). In particular, $L_{\mathrm{DPO}}$ can be modified as follows:

$$L_{\mathrm{DPO\text{-}Diffusion}} = -\mathbb{E}_{\mathcal{R}_1^0, \mathcal{R}_2^0}\left[\log \sigma\left(\beta \mathbb{E}_{\mathcal{R}_1^{1:T}, \mathcal{R}_2^{1:T}}\left[\mathrm{sgn}(\mathcal{R}_1^0, \mathcal{R}_2^0)\left(\log \frac{p_{\boldsymbol{\theta}}(\mathcal{R}_1^{0:T})}{p_{\mathrm{ref}}(\mathcal{R}_1^{0:T})} - \log \frac{p_{\boldsymbol{\theta}}(\mathcal{R}_2^{0:T})}{p_{\mathrm{ref}}(\mathcal{R}_2^{0:T})}\right)\right]\right)\right],$$

where $\mathcal{R}_1^{1:T} \sim p_{\boldsymbol{\theta}}(\mathcal{R}_1^{1:T}|\mathcal{R}_1^0)$ and $\mathcal{R}_2^{1:T} \sim p_{\boldsymbol{\theta}}(\mathcal{R}_2^{1:T}|\mathcal{R}_2^0)$.

Following Wallace et al. [46], we can utilize Jensen's inequality and convexity of function $-\log \sigma$ to derive the following upper bound of $L_{\mathrm{DPO\text{-}Diffusion}}$:

$$\tilde{L}_{\mathrm{DPO\text{-}Diffusion}} = -\mathbb{E}_{t, \mathcal{R}_1^0, \mathcal{R}_2^0, (\mathcal{R}_1^{t-1}, \mathcal{R}_1^t), (\mathcal{R}_2^{t-1}, \mathcal{R}_2^t)}\Big[$$
$$\log \sigma\left(\beta T \mathrm{sgn}(\mathcal{R}_1^0, \mathcal{R}_2^0)\left[\log \frac{p_{\boldsymbol{\theta}}(\mathcal{R}_1^{t-1}|\mathcal{R}_1^t)}{p_{\mathrm{ref}}(\mathcal{R}_1^{t-1}|\mathcal{R}_1^t)} - \log \frac{p_{\boldsymbol{\theta}}(\mathcal{R}_2^{t-1}|\mathcal{R}_2^t)}{p_{\mathrm{ref}}(\mathcal{R}_2^{t-1}|\mathcal{R}_2^t)}\right]\right)\Big],$$

where $t \sim \mathcal{U}(0, T)$, $(\mathcal{R}_1^{t-1}, \mathcal{R}_1^t)$ and $(\mathcal{R}_2^{t-1}, \mathcal{R}_2^t)$ are sampled from reverse generative process of $\mathcal{R}_1^0$ and $\mathcal{R}_2^0$, respectively, i.e., $(\mathcal{R}_1^{t-1}, \mathcal{R}_1^t) \sim p_{\boldsymbol{\theta}}(\mathcal{R}_1^{t-1}, \mathcal{R}_1^t|\mathcal{R}_1^0)$ and $(\mathcal{R}_2^{t-1}, \mathcal{R}_2^t) \sim p_{\boldsymbol{\theta}}(\mathcal{R}_2^{t-1}, \mathcal{R}_2^t|\mathcal{R}_2^0)$.

In our case, the antibodies with low energy are desired. Thus, we define the reward $r(\cdot)$ as $-\mathcal{E}(\cdot)/\mathcal{T}$, where $\mathcal{E}(\cdot)$ is the energy function and $\mathcal{T}$ is the temperature. Different from the text-to-image generation where the (latent) reward is assigned to a complete image instead of a pixel [46], we know more fine-grained credit assignment. Specifically, it is known that $\mathcal{E}(\mathcal{R}^0) = \sum_{j=n+1}^{n+m} \mathcal{E}(\mathcal{R}^0[j])$, i.e., the energy of an antibody is the summation of the energy of its amino acids [4]. Thus the preference can be measured **at the residue level** instead of the entire CDR level. Besides, we have $\log p_{\boldsymbol{\theta}}(\mathcal{R}^{t-1}|\mathcal{R}^t) = \sum_{j=n+1}^{n+m} \log p_{\boldsymbol{\theta}}(\mathcal{R}^{t-1}[j]|\mathcal{R}^t)$, which is a common assumption of diffusion models. Thus we can derive a residue-level DPO-Diffusion loss:

$$L_{\mathrm{residue\text{-}DPO\text{-}Diffusion}} = -\mathbb{E}_{t, \mathcal{R}_1^0, \mathcal{R}_2^0, (\mathcal{R}_1^{t-1}, \mathcal{R}_1^t), (\mathcal{R}_2^{t-1}, \mathcal{R}_2^t)}\Big[$$
$$\log \sigma\left(\beta T \sum_{j=n+1}^{n+m} \mathrm{sgn}(\mathcal{R}_1^0[j], \mathcal{R}_2^0[j])\left[\log \frac{p_{\boldsymbol{\theta}}(\mathcal{R}_1^{t-1}[j]|\mathcal{R}_1^t)}{p_{\mathrm{ref}}(\mathcal{R}_1^{t-1}[j]|\mathcal{R}_1^t)} - \log \frac{p_{\boldsymbol{\theta}}(\mathcal{R}_2^{t-1}[j]|\mathcal{R}_2^t)}{p_{\mathrm{ref}}(\mathcal{R}_2^{t-1}[j]|\mathcal{R}_2^t)}\right]\right)\Big].$$

Thus, by Jensen's inequality and the convexity of $-\log \sigma$, we can further derive $\tilde{L}_{\text{residue-DPO-Diffusion}}$, which is an upper bound of $L_{\text{residue-DPO-Diffusion}}$:

$$\tilde{L}_{\text{residue-DPO-Diffusion}} = -\mathbb{E}_{t,\mathcal{R}_1^0,\mathcal{R}_2^0,(\mathcal{R}_1^{t-1},\mathcal{R}_1^t),(\mathcal{R}_2^{t-1},\mathcal{R}_2^t)}\Bigg[$$

$$\sum_{j=n+1}^{n+m} \log \sigma \left( \beta T \operatorname{sgn}(\mathcal{R}_1^0[j],\mathcal{R}_2^0[j]) \left[ \log \frac{p_{\boldsymbol{\theta}}(\mathcal{R}_1^{t-1}[j]|\mathcal{R}_1^t)}{p_{\text{ref}}(\mathcal{R}_1^{t-1}[j]|\mathcal{R}_1^t)} - \log \frac{p_{\boldsymbol{\theta}}(\mathcal{R}_2^{t-1}[j]|\mathcal{R}_2^t)}{p_{\text{ref}}(\mathcal{R}_2^{t-1}[j]|\mathcal{R}_2^t)} \right] \right) \Bigg].$$

The gradients of $\tilde{L}_{\text{DPO-Diffusion}}$ and $\tilde{L}_{\text{residue-DPO-Diffusion}}$ w.r.t the parameters $\boldsymbol{\theta}$ can be written as:

$$\nabla_{\boldsymbol{\theta}}\tilde{L}_{\text{DPO-Diffusion}} = -\beta T \mathbb{E}_{t,\mathcal{R}_1^0,\mathcal{R}_2^0,(\mathcal{R}_1^{t-1},\mathcal{R}_1^t),(\mathcal{R}_2^{t-1},\mathcal{R}_2^t)}\Big[ \sum_{j=n+1}^{n+m} \operatorname{sgn}(\mathcal{R}_1^0,\mathcal{R}_2^0)$$

$$\cdot \sigma(\hat{r}(\mathcal{R}_2^0) - \hat{r}(\mathcal{R}_1^0))\Big( \nabla_{\boldsymbol{\theta}}\log p_{\boldsymbol{\theta}}(\mathcal{R}_1^{t-1}[j]|\mathcal{R}_1^t) - \nabla_{\boldsymbol{\theta}}\log p_{\boldsymbol{\theta}}(\mathcal{R}_2^{t-1}[j]|\mathcal{R}_2^t)\Big)\Big],$$

and

$$\nabla_{\boldsymbol{\theta}}\tilde{L}_{\text{residue-DPO-Diffusion}} = -\beta T \mathbb{E}_{t,\mathcal{R}_1^0,\mathcal{R}_2^0,(\mathcal{R}_1^{t-1},\mathcal{R}_1^t),(\mathcal{R}_2^{t-1},\mathcal{R}_2^t)}\Big[ \sum_{j=n+1}^{n+m} \operatorname{sgn}(\mathcal{R}_1^0[j],\mathcal{R}_2^0[j])$$

$$\cdot \sigma(\hat{r}(\mathcal{R}_2^0[j]) - \hat{r}(\mathcal{R}_1^0[j]))\Big( \nabla_{\boldsymbol{\theta}}\log p_{\boldsymbol{\theta}}(\mathcal{R}_1^{t-1}[j]|\mathcal{R}_1^t) - \nabla_{\boldsymbol{\theta}}\log p_{\boldsymbol{\theta}}(\mathcal{R}_2^{t-1}[j]|\mathcal{R}_2^t)\Big)\Big],$$

where $\hat{r}(\cdot) := \log(p_{\boldsymbol{\theta}}(\cdot)/p_{\text{ref}}(\cdot))$, which can be viewed as the estimated reward by current policy $p_{\boldsymbol{\theta}}$.

We can see that $\nabla_{\boldsymbol{\theta}}\tilde{L}_{\text{DPO-Diffusion}}$ actually reweight $\nabla_{\boldsymbol{\theta}}\log p_{\boldsymbol{\theta}}(\mathcal{R}^{t-1}[j]|\mathcal{R}^t)$ with the estimated reward of the complete antibody while $\nabla_{\boldsymbol{\theta}}\tilde{L}_{\text{residue-DPO-Diffusion}}$ does this with the estimated reward of the amino acid itself. In this case, $\nabla_{\boldsymbol{\theta}}\tilde{L}_{\text{DPO-Diffusion}}$ will increase (resp. decrease) the likelihood of all amino acids of the "winning" sample (resp. "losing") at the same rate, which may mislead the optimization direction. In contrast, $\nabla_{\boldsymbol{\theta}}\tilde{L}_{\text{residue-DPO-Diffusion}}$ does not have this issue and can fully utilize the residue-level signals from estimated reward to effectively optimize antibodies.

We further approximate the objective $\tilde{L}_{\text{residue-DPO-Diffusion}}$ by sampling from the forward diffusion process $q$ instead of the reverse generative process $p_{\boldsymbol{\theta}}$ to achieve diffusion-like efficient training. With further replacing $\log \frac{p_{\boldsymbol{\theta}}}{p_{\text{ref}}}$ with $-\log \frac{q}{p_{\boldsymbol{\theta}}} + \log \frac{p_{\text{ref}}}{q}$ which is exactly $-\mathbb{D}_{KL}(q\|p_{\boldsymbol{\theta}}) + \mathbb{D}_{KL}(q\|p_{\text{ref}})$ when taking expectation with respect to $q$, we can derive the final loss for fine-tuning the diffusion model as follows:

$$L_{\text{ABDPO}} = -\mathbb{E}_{t,\mathcal{R}_1^0,\mathcal{R}_2^0,(\mathcal{R}_1^{t-1},\mathcal{R}_1^t),(\mathcal{R}_2^{t-1},\mathcal{R}_2^t)}\Big[ \sum_{j=n+1}^{n+m} \log \sigma\Big( -\beta T \operatorname{sgn}(\mathcal{R}_1^0[j],\mathcal{R}_2^0[j])$$

$$\cdot \big\{ \mathbb{D}_{\text{KL},1}^t(q\|p_{\boldsymbol{\theta}})[j] - \mathbb{D}_{\text{KL},1}^t(q\|p_{\text{ref}})[j] - \mathbb{D}_{\text{KL},2}^t(q\|p_{\boldsymbol{\theta}})[j] + \mathbb{D}_{\text{KL},2}^t(q\|p_{\text{ref}})[j] \big\} \Big)\Big], \qquad (8)$$

where $\mathcal{R}_1^0, \mathcal{R}_2^0 \sim p_{\boldsymbol{\theta}}(\mathcal{R})$, $(\mathcal{R}_1^{t-1},\mathcal{R}_1^t)$ and $(\mathcal{R}_2^{t-1},\mathcal{R}_2^t)$ are sampled from forward diffusion process of $\mathcal{R}_1^0$ and $\mathcal{R}_2^0$, respectively, which can be much more efficient than the reverse generative process that involves hundreds of model forward estimation. Here we use $\mathbb{D}_{\text{KL},1}^t(q\|p_{\boldsymbol{\theta}})[j]$ to denote $\mathbb{D}_{\text{KL}}(q(\mathcal{R}_1^{t-1}[j]|\mathcal{R}^t,\mathcal{R}^0)\|p_{\boldsymbol{\theta}}(\mathcal{R}_1^{t-1}[j]|\mathcal{R}^0))$. Similar for $\mathbb{D}_{\text{KL},1}^t(q\|p_{\text{ref}})[j]$, $\mathbb{D}_{\text{KL},2}^t(q\|p_{\boldsymbol{\theta}})[j]$, and $\mathbb{D}_{\text{KL},2}^t(q\|p_{\text{ref}})[j]$. These KL divergence can be estimated as in Eqs. (5) to (7).

## 3.3 Energy Decomposition and Conflict Mitigation

The energy usually consists of different types, such as attraction and repulsion. Empirically, direct optimization on single energy will lead to some undesired "shortcuts". Specifically, in some cases, repulsion dominates the energy of the antibody so the model will push antibodies as far from the antigen as possible to decrease the repulsion during optimization, and finally fall into a bad local minima. This effectively reduces the repulsion, but also completely eliminates the attraction between antibodies and antigens, which seriously impairs the functionality of the antibody. This motivates us to explicitly express the energy with several distinct terms and then control the optimization process towards our preference.

Inspired by Yu et al. [51], we utilize "gradient surgery" to alleviate interference between different types of energy during energy preference optimization. More specifically, we have $\mathcal{E}(\cdot) = \sum_{v=1}^V w_v \mathcal{E}_v(\cdot)$, where $V$ is the number of types of energy, and $w_v$ is a constant weight for the $v$-th kind of energy. For each type of energy $\mathcal{E}_v(\cdot)$, we compute its corresponding energy preference gradient $\nabla_{\boldsymbol{\theta}} L_v$ as

Table 1: Summary of AAR, RMSD, CDR $E_{\text{total}}$, CDR-Ag $\Delta G$ (kcal/mol), pLL, PHR, and $N_{\text{success}}$ of antibodies designed by our model and baselines. ($\downarrow$) / ($\uparrow$) denotes a smaller / larger number is better.

| Methods | AAR ($\uparrow$) | RMSD ($\downarrow$) | CDR $E_{\text{total}}$ ($\downarrow$) | CDR-Ag $\Delta G$ ($\downarrow$) | pLL ($\uparrow$) | PHR ($\downarrow$) | $N_{\text{success}}$ ($\uparrow$) |
|---|---|---|---|---|---|---|---|
| HERN | 32.38% | 9.18 | 10887.77 | 2095.88 | -2.02 | 40.46% | 0 |
| MEAN | 36.20% | **1.69** | 7162.65 | 1041.43 | **-1.79** | **30.62%** | 0 |
| dyMEAN | **40.04%** | 1.82 | 3782.67 | 1730.06 | -1.82 | 43.72% | 0 |
| DiffAb | 34.92% | 1.92 | 1729.51 | 1297.25 | -2.10 | 41.27% | 0 |
| ABDPO | 31.25% | 1.98 | **629.44** | **307.56** | -2.18 | 69.67% | **9** |
| ABDPO+ | 36.27% | 2.01 | 1106.48 | 637.62 | -2.00 | 44.21% | 5 |

Eq. (8), and then alter the gradient by projecting it onto the normal plane of the other gradients (in a random order) if they have conflicts. This process works as follows:

$$\nabla_{\boldsymbol{\theta}} L_v \leftarrow \nabla_{\boldsymbol{\theta}} L_v - \frac{\min\left(\nabla_{\boldsymbol{\theta}} L_v^\top \nabla_{\boldsymbol{\theta}} L_u, 0\right)}{\left\|\nabla_{\boldsymbol{\theta}} L_u\right\|^2} \nabla_{\boldsymbol{\theta}} L_u, \tag{9}$$

where $v \in \{1, \ldots, V\}$ and $u = \texttt{Shuffle}(1, \ldots, V)$.

## 4 Experiments

### 4.1 Experimental Setup

**Dataset Curation**    To pre-train the diffusion model for antibody generation, we use the Structural Antibody Database [SAbDab, 13] under IMGT [34] scheme as the dataset. We collected antigen-antibody complexes with both heavy and light chains and protein antigens and discarded the duplicate data with the same CDR-L3 and CDR-H3 sequence. The remaining complexes are used to cluster via MMseqs2 [44] with 40% sequence similarity as the threshold based on the CDR-H3 sequence of each complex. We then select the clusters that do not contain complexes in RAbD benchmark [1] and split the complexes into training and validation sets with a ratio of 9:1 (1786 and 193 complexes respectively). Specifically, the validation set is composed of clusters that only contain one complex. The test set consists of 55 eligible complexes from the RAbD benchmark (details in Appendix D.2).

For the synthetic data used in ABDPO fine-tuning, 10,112 samples are randomly sampled for each antigen-antibody complex in the test set using the aforementioned pre-trained diffusion model. Then, we use pyRosetta [9] to apply the side-chain packing for these samples.

**Preference Definition**    To apply ABDPO, we need to build the preference dataset and construct the "winning" and "losing" pair. The accurate relationship between preferences based on *in silico* with wet-lab experimental results is a scientific issue that remains unresolved, with a wide range of opinions. ABDPO's solution to this open question is to provide a generic framework that allows for arbitrary definitions and combinations of preferences to satisfy various requirements in antibody design.

To demonstrate the effectiveness of ABDPO, we define the preferences as lower total energy and lower binding energy. The two energies are defined on residue level, specifically, **(1)** $\text{Res}_{\text{CDR}} E_{\text{total}}$ is the total energy of each residue within the designed CDR, and is used to represent the overall rationality of the corresponding residue; **(2)** $\text{Res}_{\text{CDR}}$-Ag $\Delta G$ is the interaction energy between each designed CDR residue and the target antigen, representing the functionality of the corresponding residue. $\text{Res}_{\text{CDR}}$-Ag $\Delta G$ is further decomposed into **(2.1)** $\text{Res}_{\text{CDR}}$-Ag $E_{\text{nonRep}}$, the sum of the interaction energies except repulsion between the designed CDR residue and the antigen, and **(2.2)** $\text{Res}_{\text{CDR}}$-Ag $E_{\text{Rep}}$, the repulsion energy between the design CDR residue and the antigen.

As a generic framework, ABDPO also supports non-energy-based preferences. To verify this, we demonstrate an advanced version named ABDPO+. ABDPO+ incorporates two additional preferences: pseudo log-likelihood (pLL) from AntiBERTy [42] and the percent of hydrophobicity residues (PHR). Different from the previously mentioned energy-based preferences, pLL and PHR are defined on the whole CDR level. For pLL, a higher value is considered better and is designated as "winning", conversely; for PHR, a lower value is preferable.

**Baselines**    We compare our model with various representative antibody sequence-structure co-design baselines. **HERN** [21] designs sequences of antibodies autoregressively with the iterative refinement

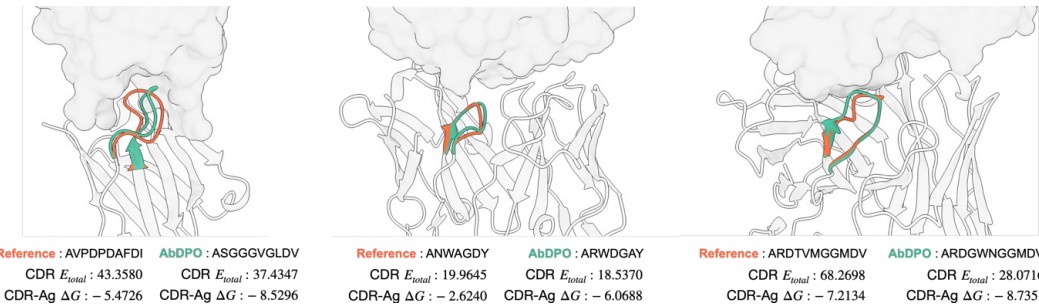

Figure 3: Visualization of reference antibodies in RAbD and antibodies designed by ABDPO given specific antigens (PDB ID: **1iqd** (left), **1ic7** (middle), and **2dd8** (right)). The unit of energy annotated is kcal/mol and omitted here for brevity.

of structures; **MEAN** [29] generates sequences and structures of antibodies via a progress full-shot scheme; **dyMEAN** [30] designs antibodies sequences and structures with full-atom modeling; **DiffAb** [36] models antibody distributions with a diffusion model that considers the amino acid type, $C_\alpha$ positions and side-chain orientations, which is a more rigorous *generative* model than the above baselines. The side-chain atoms are packed by pyRosetta. For **dyMEAN**, we **(1)** provide the ground-truth framework structure as input like other methods, **(2)** only use its generated backbones and pack the side-chain atoms by pyRosetta for a more fair comparison.

**Evaluation**    Following the previous studies, we preliminarily evaluate the generated sequence and structure with AAR and $C\alpha$ RMSD. Besides, we carry out a series of more reasonable metrics. We utilize the preferences aforementioned to evaluate the designed antibodies from multiple perspectives, but at the whole CDR level. Specifically, **(1)** CDR $E_{\text{total}}$, the total energy of the designed CDR, is utilized to evaluate the rationality by aggregating all $\text{Res}_{\text{CDR}}$ $E_{\text{total}}$ of residues within the CDR; **(2)** CDR-Ag $\Delta G$ denotes the difference in total energy between the bound state and the unbound state of the CDR and antigen, which is calculated to evaluate the functionality. PHR and pLL remain the same definition as above. All methods are able to generate multiple antibodies for a specific antigen (a randomized version of MEAN, rand-MEAN, is used here). We employ each method to design 192 antibodies for each complex, and we report the mean metrics across all 55 complexes. We further report the number of successfully designed antibody-antigen complexes, $N_{\text{success}}$, to evaluate their rationality and functionality comprehensively. The design for an antibody-antigen complex is considered as "successful" when at least one generated sample holds energies close to or lower than the natural one, i.e., for both of the two energy types, $E_{\text{generated}} < E_{\text{natural}} + \text{std}(E_{\text{natural}}^{\text{all-complexes}})$.

## 4.2   Main Results

We report the evaluation metrics in Tab. 1. As the results show, ABDPO performs significantly superior to other antibody sequence-structure co-design methods in the two energy-based metrics, CDR $E_{\text{total}}$ and CDR-Ag $\Delta G$, while maintaining the AAR and RMSD. With the two additional preferences, ABDPO+ avoids the expense of the increased PHR while achieving better performance than DiffAb in remaining metrics (even surpassing DiffAb in AAR). This demonstrates the effectiveness and compatibility of ABDPO in terms of optimizing multi-objectives simultaneously. We have also provided the detailed evaluation results for each complex in Appendix E.2.

We do not consider AAR and RMSD as the main reference evaluation metrics as their inadequacy (refer to Appendix A for more details). With the new evaluation methods, issues that used to be hidden by AAR and RMSD are exposed. It is observed that structural clashes can not be avoided completely in any method, resulting in the high energy values of generated antibodies, even for ABDPO and ABDPO+. The structural clashes between CDR and the antigen finally lead to the unreasonable high CDR-Ag $\Delta G$. However, the primary goal in antibody design is to generate at least one effective antibody. Given the complexity of protein interactions, it is not plausible that every generated antibody will yield effectiveness. Therefore, $N_{\text{success}}$ is a more valuable metric. ABDPO and ABDPO+ are the only two to achieve successful cases, with 9 and 5 successful cases out of 55 complexes, respectively. Following this concept, we also rank the designed antibodies for each complex by a uniform strategy (see Appendix D.3), calculate the metrics of the highest-ranked design for each complex, and report the mean metrics across the 55 complexes (see Appendix E.1). Notably, ABDPO is the only method that achieves CDR-Ag $\Delta G$ lower than 0.

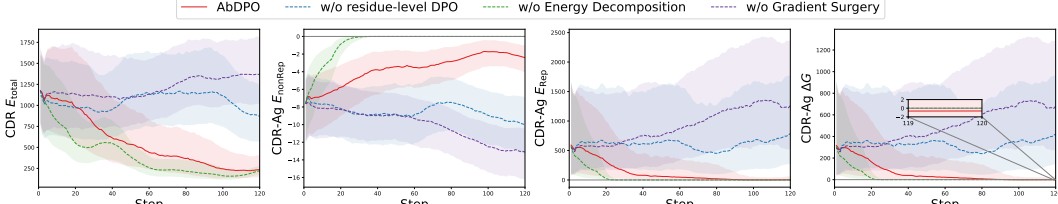

Figure 4: Changes of median CDR $E_{\text{total}}$, $E_{\text{nonRep}}$, $E_{\text{Rep}}$, and CDR-Ag $\Delta G$ (kcal/mol) over-optimization steps, shaded to indicate interquartile range (from 25-th percentile to 75-th percentile).

We also visualize three cases (PDB ID: 1iqd, 1ic7, and 2dd8) in Fig. 3. It is shown that ABDPO can design CDRs with both fewer clashes and proper relative spatial positions towards the antigens, and even better energy performance than that of natural antibodies.

We conduct another two experiments to demonstrate further the generality of ABDPO: **(1)** directly incorporate auxiliary training losses for those properties of which gradients are computable; **(2)** introduce energy minimization before energy calculation, which is more in line with the real workflow. ABDPO shows consistent performance and demonstrates its generality. Please refer to Appendix F for related details.

### 4.3 Ablation Studies

Our approach comprises three main novel designs, including residue-level direct energy-based preference optimization, energy decomposition, and conflict mitigation by gradient surgery. Thus we perform comprehensive ablation studies to verify our hypothesis on the effects of each respective design component. Here we take the experiment on one complex (PDB ID: 1a14) as the example. Here, we apply more fine-tuning steps and additionally introduce $E_{\text{nonRep}}$ (aggregation of $\text{Res}_{\text{CDR}}$-Ag $E_{\text{nonRep}}$ within the designed CDR), $E_{\text{Rep}}$ (aggregation of $\text{Res}_{\text{CDR}}$-Ag $E_{\text{Rep}}$) for a more obvious and detailed comparison. More cases of ablation studies can be found in Appendix G.

**Effects of Residue-level Energy Preference Optimization** We hypothesize that residue-level DPO leads to more explicit and intuitive gradients that can promote effectiveness and efficiency compared with the vanilla DPO [46] as the analysis in Sec. 3.2. To validate this, we compare ABDPO with its counterpart with the CDR-level preference instead of residue-level. As Fig. 4 shows, regarding the counterpart (blue dotted line), the changes in all metrics are not obvious, while almost all metrics rapidly converge to an ideal state in ABDPO (red line). This demonstrated the effects of residue-level energy preference in improving the optimization efficiency.

**Effects of Energy Decomposition** In generated antibodies, the huge repulsion caused by clashes accounts for the majority of the two types of energy. This prevents us from using the $\Delta G$ as an optimization objective directly as the model is allowed to minimize repulsion by keeping antibodies away from antigens, quickly reducing the energies. To verify this, we compared ABDPO with a version that directly optimize $\Delta G$. As shown in Fig. 4, without energy decomposition (green dashed line), both $E_{\text{Rep}}$ and $E_{\text{nonRep}}$ quickly diminish to 0, indicating that there is no interaction between the generated antibodies and antigens. Conversely, ABDPO (red line) can minimize $E_{\text{Rep}}$ to 0 while maintaining $E_{\text{nonRep}}$, which means the interactions are preserved.

**Effects of Gradient Surgery** To show the effectiveness of gradient surgery in mitigating conflicts when optimizing multiple objectives, we compare ABDPO and its counterpart without gradient surgery. As Fig. 4 shows, the counterpart (purple dashed line) can only slightly optimize CDR-Ag $E_{\text{nonRep}}$ but incurs strong repulsion (i.e., $E_{\text{Rep}}$), learning to irrational structures. ABDPO (red line) can converge to a state where CDR $E_{\text{total}}$ and $E_{\text{Rep}}$ achieve a conspicuously low point, suggesting the generated sequences and structures are stable, and $E_{\text{nonRep}}$ is still significantly less than zero, showing that considerable binding affinity is kept.

**Comparison with Supervised Fine-tuning** Supervised Fine-tuning (SFT) can be an alternative way of generating antibodies with lower energy. For SFT, we first select the top 10% high-quality samples from ABDPO training data on a complex (PDB ID: 1a14). We fine-tune the diffusion model under the same settings as ABDPO. Results in Tab. 2 show that SFT only marginally surpasses the pre-trained diffusion model, and ABDPO performs significantly superior to SFT. We attribute the performance of ABDPO to the preference optimization scheme and the fine-grained residue-level energy rather than the entire CDR.

Table 2: Comparison of ABDPO and supervised fine-tuning (SFT) on 1a14.

| Methods | CDR $E_{\text{total}}$ ($\downarrow$) | | CDR-Ag $\Delta G$ ($\downarrow$) | |
| --- | --- | --- | --- | --- |
| | Avg. | Med. | Avg. | Med. |
| DiffAb | 1314.20 | 1133.36 | 534.21 | 248.28 |
| DiffAb$_{\text{SFT}}$ | 1053.82 | 869.37 | 374.27 | 144.25 |
| ABDPO | **336.02** | **226.25** | **88.64** | **0.10** |

## 5 Conclusions

In this work, we rethink antibody sequence-structure co-design through the lens of energy and propose ABDPO for designing antibodies meeting multi-objectives like rationality and functionality. The introduction of direct energy-based preference optimization along with energy decomposition and conflict mitigation by gradient surgery shows promising results in generating antibodies with low energy and high binding affinity. With ABDPO, existing computing software and domain knowledge can be easily combined with deep learning techniques, jointly facilitating the development of antibody design. Limitations and future work are discussed in Appendix H.

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

# A    Motivation for Choosing Energy as Evaluation

There are many inadequacies in using AAR and RMSD as the main evaluation metrics in AI-based antibody design. Antibody design is a typical function-oriented protein design task, necessitating a more fine-grained measure of discrepancy compared to general protein design tasks. Especially when the part of the antibody to be designed and evaluated, CDR-H3, is usually shorter, more precise evaluation becomes particularly important.

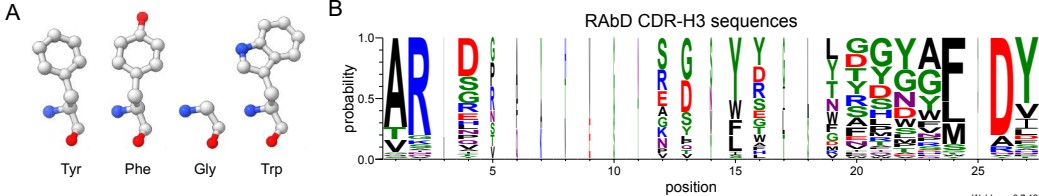

Figure 5: **A**: Tyr (Y) and Phe (F) differ by only one oxygen atom. In contrast, there is a substantial difference between Gly (G) and Trp (W). Gly lacks a side chain, whereas Trp possesses the largest side chain of all amino acids. **B**: the visualization of the frequency of occurrence of each amino acid at various positions in RAbD CDR-H3 sequences. The sequences are initially aligned using MAFFT [26] and subsequently visualized with WebLogo [11]. The width of each column corresponds to the frequency of occurrence at that position.

For AAR, there are two main limitations in measuring the similarity between the generated sequence and the reference sequence. The first limitation is located in measuring the difference in different incorrect recoveries. Among the 20 common amino acids, some have high similarity between them, such as Tyr and Phe, while others have significant differences, such as Gly and Trp (Fig. 5A). When an amino acid in CDR is erroneously recovered to different amino acids, their impact will also vary. However, AAR does not differentiate between these different types of errors, only identifying them as "incorrect".

A further, more serious issue is that AAR is easily hacked. Although the CDR region is often considered hypervariable, a mild conservatism in sequence still exists (Fig. 5B), which allows the model to obtain satisfactory AAR using a simple but incorrect way - directly generating the amino acids with the highest probability of occurrence at each position, while ignoring the condition of the given antigen which is extremely harmful to the specificity of antibodies. We made a simple attempt by simply counting the amino acids with the highest frequency of occurrence at various positions in all samples in SAbDab, and then composing them into a CDR-H3 sequence, which looks roughly like "$\text{ARD} + \text{rand}(Y, G)* + \text{FDY}$", achieving an AAR of **38.77**% on the RAbD dataset.

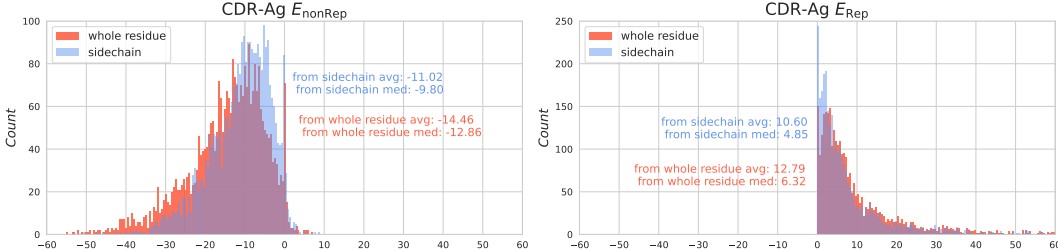

Figure 6: The distribution of CDR-Ag $E_{\text{nonRep}}$ (left) and CDR-Ag $E_{\text{Rep}}$ (right) formed by the whole CDR atoms (colored in red) and solely by CDR side-chain atoms (colored in blue) among SAbDab dataset.

While RMSD fails to measure the discrepancies on side-chain atoms, in general, the calculation of RMSD focuses on the alpha carbon atom or the four backbone atoms due to their stable existence in any type of amino acid and thus ignores the side-chain atoms. However, side-chain atoms in the CDR region are extremely important as they contribute to most of the interactions between the CDR and the antigen. Our analyses on the SAbDab dataset also prove the importance of the side chain in CDR-Antigen interaction in terms of energy. As shown in Fig. 6, the distribution of energies formed

by the whole residues in CDR is colored in red while the distribution of energies formed only by side-chain atoms of CDR is colored in blue. The interaction energy formed by side-chain atoms accounts for the vast majority of the total interaction energy in both types of energy.

The above reasons have led us to abandon AAR and RMSD as learning objectives and evaluation metrics, and instead use energy as our goal. Energy can simultaneously consider the relationship between structure and sequence, distinguish different generation results in more detail, and importantly, reflect the rationality and functionality of antibodies in a more fundamental way. Despite the various shortcomings of AAR and RMSD, we have demonstrated that the antibodies generated by ABDPO achieve lower AAR and comparable RMSD compared to those generated by other methods. However, in practice, ABDPO-generated antibodies exhibit distinct binding patterns to antigens, differing from reference antibodies, and demonstrate significantly better energy performance than those produced by other methods. This further highlights the inadequacies of using AAR and RMSD as evaluation metrics in antibody design tasks, exposing their vulnerability to being "hacked".

## B  Energy Calculation

In ABDPO, we conduct the calculation on $\text{Res}_{\text{CDR}}\ E_{\text{total}}$ at residue level, and a more fine-grained calculation on the two functionality-associated energies at the sub-residue level. We use Rosetta to calculate all types of energies in this paper.

We denote the residue with the index $i$ in the antibody-antigen complex as $A_i$, then $A_i^{sc}$ and $A_i^{bb}$ represent the **side chain** and **backbone** of the residue respectively.

For the energies in the proposed preference, we describe the function for energies of a **S**ingle residue as ES, and $\text{ES}_{\text{total}}$ is the sum of all types of energy with the default weight in REF15 [4]. The function for interaction energies between **P**aired residues is described as EP, which consists of six different energy types: $\text{EP}_{\text{hbond}}$, $\text{EP}_{\text{att}}$, $\text{EP}_{\text{rep}}$, $\text{EP}_{\text{sol}}$, $\text{EP}_{\text{elec}}$, and $\text{EP}_{\text{lk}}$.

Following the settings previously mentioned in Sec. 3.1, the indices of residues within the CDR-H3 range from $n+1$ to $n+m$, and the indices of residues within the antigen range from $g+1$ to $g+k$. Then, for the CDR residue with the index $j$, the three types of energy are defined as:

$$\text{Res}_{\text{CDR}}\ E_{\text{total}}^{j} = \text{ES}_{\text{total}}(A_j), \tag{10}$$

$$\text{Res}_{\text{CDR}}\text{-Ag}\ E_{\text{nonRep}}^{j} = \sum_{i=g+1}^{g+k} \sum_{e \in \{\text{hbond,att,sol,elec,lk}\}} \left( \text{EP}_{\text{e}}(A_j^{sc}, A_i^{sc}) + \text{EP}_{\text{e}}(A_j^{sc}, A_i^{bb}) \right), \tag{11}$$

$$\text{Res}_{\text{CDR}}\text{-Ag}\ E_{\text{Rep}}^{j} = \sum_{i=g+1}^{g+k} \left( \text{EP}_{\text{rep}}(A_j^{sc}, A_i^{sc}) + \text{EP}_{\text{rep}}(A_j^{sc}, A_i^{bb}) \right.$$
$$\left. + 2 \times \text{EP}_{\text{rep}}(A_j^{bb}, A_i^{sc}) + 2 \times \text{EP}_{\text{rep}}(A_j^{bb}, A_i^{bb}) \right). \tag{12}$$

It can be observed from Eqs. (11) and (12) that the two functionality-associated energies, namely $\text{Res}_{\text{CDR}}\text{-Ag}\ E_{\text{nonRep}}$ and $\text{Res}_{\text{CDR}}\text{-Ag}\ E_{\text{Rep}}$, which collectively describe the interaction energy between CDR and the antigen, are computed at the level of side-chain and backbone. $\text{Res}_{\text{CDR}}\text{-Ag}\ E_{\text{nonRep}}$ is only calculated on the interactions caused by the side-chain atoms in the CDR-H3 region, while $\text{Res}_{\text{CDR}}\text{-Ag}\ E_{\text{Rep}}$ assigns a greater cost to the repulsions caused by the backbone atoms in the CDR-H3 region. This modification is carried out according to the fact that the side-chain atoms contribute the vast majority of energy to the interaction between CDR-H3 and antigens (Fig. 6), and $E_{\text{nonRep}}$ exhibits a benefit in interactions, while $E_{\text{Rep}}$ could be regarded as a cost.

The fine-grained calculation of $\text{Res}_{\text{CDR}}\text{-Ag}\ E_{\text{nonRep}}$ and $\text{Res}_{\text{CDR}}\text{-Ag}\ E_{\text{Rep}}$ is indispensable. Without the fine-grained calculation, the model tends to generate poly-G CDR-H3 sequences, such as "GGGGGGGGGGGG" for any given antigen and the rest of the antibody. The most likely reason for this is that G, Glycine, can maximize the reduction of clashes and gain satisfactory CDR $E_{\text{total}}$ and $\text{Res}_{\text{CDR}}\text{-Ag}\ E_{\text{Rep}}$ as it doesn't contain side chain and simultaneously form a weak attraction to the antigen solely relying on its backbone atoms.

We emphasize that the two functionality-associated energies, $\text{Res}_{\text{CDR}}\text{-Ag}\ E_{\text{nonRep}}$ and $\text{Res}_{\text{CDR}}\text{-Ag}\ E_{\text{Rep}}$ are calculated exclusively at the sub-residue level when serving as the determination of preference in

guiding the direct energy-based preference optimization process. However, when these energies are used as evaluation metrics, they are calculated at the residue level, in which the greater cost to the repulsions attributed to the backbone atoms is negated.

## C  Theoretical Justification

In this section, we show the detailed mathematical derivations of formulas in Sec. 3.2. Although many of them are similar to Rafailov et al. [41], we still present them in detail for the sake of completeness. Besides, we will also present the details of preference data generation.

First, we will show the derivation of the optimal solution of the KL-constrained reward-maximization objective, i.e., $\max_{p_\theta} \mathbb{E}_{\mathcal{R}^0 \sim p_\theta}[r(\mathcal{R}^0)] - \beta \mathbb{D}_{\mathrm{KL}}(p_\theta(\mathcal{R}^0) \| p_{\mathrm{ref}}(\mathcal{R}^0))$ as follows:

$$
\max_{p_\theta} \mathbb{E}_{\mathcal{R}^0 \sim p_\theta}[r(\mathcal{R}^0)] - \beta \mathbb{D}_{\mathrm{KL}}(p_\theta(\mathcal{R}^0) \| p_{\mathrm{ref}}(\mathcal{R}^0))
$$

$$
= \max_{p_\theta} \mathbb{E}_{\mathcal{R}^0 \sim p_\theta} \left[ r(\mathcal{R}^0) - \beta \log \frac{p_\theta(\mathcal{R}^0)}{p_{\mathrm{ref}}(\mathcal{R}^0)} \right]
$$

$$
= \min_{p_\theta} \mathbb{E}_{\mathcal{R}^0 \sim p_\theta} \left[ \log \frac{p_\theta(\mathcal{R}^0)}{p_{\mathrm{ref}}(\mathcal{R}^0)} - \frac{1}{\beta} r(\mathcal{R}^0) \right]
$$

$$
= \min_{p_\theta} \mathbb{E}_{\mathcal{R}^0 \sim p_\theta} \left[ \log \frac{p_\theta(\mathcal{R}^0)}{\frac{1}{Z} p_{\mathrm{ref}}(\mathcal{R}^0) \exp\left(\frac{1}{\beta} r(\mathcal{R}^0)\right)} - \log Z \right]
$$

where $Z$ is the partition function that does not involve the model being trained, i.e., $p_\theta$. And we can define

$$
p^*(\mathcal{R}^0) := \frac{1}{Z} p_{\mathrm{ref}}(\mathcal{R}^0) \exp\left(\frac{1}{\beta} r(\mathcal{R}^0)\right).
$$

With this, we can now arrive at

$$
\min_{p_\theta} \mathbb{E}_{\mathcal{R}^0 \sim p_\theta} \left[ \log \frac{p_\theta(\mathcal{R}^0)}{p^*(\mathcal{R}^0)} \right] - \log Z
$$

$$
= \min_{p_\theta} \mathbb{E}_{\mathcal{R}^0 \sim p_\theta}[\mathbb{D}_{\mathrm{KL}}(p_\theta \| p^*)] + Z
$$

Since $Z$ does not depend on $p_\theta$, we can directly drop it. According to Gibb's inequality that KL-divergence is minimized at 0 if and only if the two distributions are identical. Hence we arrive at the optimum as follows:

$$
p_{\theta^*}(\mathcal{R}^0) = p^*(\mathcal{R}^0) = \frac{1}{Z} p_{\mathrm{ref}}(\mathcal{R}^0) \exp\left(\frac{1}{\beta} r(\mathcal{R}^0)\right). \tag{13}
$$

Then we will show that the objective that maximizes likelihood on preference data sampled from $p(\mathcal{R}_1^0 \succ \mathcal{R}_2^0) = \sigma(r(\mathcal{R}_1^0) - r(\mathcal{R}_2^0))$, which is exactly $L_{\mathrm{DPO}}$, leads to the same optimal solution. For this, we need to express the pre-defined reward $r(\cdot)$ with the optimal policy $p^*$:

$$
r(\mathcal{R}^0) = \beta \log \frac{p^*(\mathcal{R}^0)}{p_{\mathrm{ref}}(\mathcal{R}^0)} + Z
$$

The we plugin the expression of $r(\cdot)$ into $p(\mathcal{R}_1^0 \succ \mathcal{R}_2^0) = \sigma(r(\mathcal{R}_1^0) - r(\mathcal{R}_2^0))$ as follows:

$$
p(\mathcal{R}_1^0 \succ \mathcal{R}_2^0) = \sigma(r(\mathcal{R}_1^0) - r(\mathcal{R}_2^0))
$$

$$
= \sigma\left( \beta \log \frac{p^*(\mathcal{R}_1^0)}{p_{\mathrm{ref}}\mathcal{R}_1^0)} - \beta \log \frac{p^*(\mathcal{R}_2^0)}{p_{\mathrm{ref}}(\mathcal{R}_2^0)} \right),
$$

where $Z$ is canceled out. For brevity, we use the following notation for brevity:

$$
p_\theta(\mathcal{R}_1^0 \succ \mathcal{R}_2^0) = \sigma\left( \beta \log \frac{p_\theta(\mathcal{R}_1^0)}{p_{\mathrm{ref}}(\mathcal{R}_1^0)} - \beta \log \frac{p_\theta(\mathcal{R}_2^0)}{p_{\mathrm{ref}}(\mathcal{R}_2^0)} \right).
$$

With this, we have

$$
\min_{p_\theta} L_{\mathrm{DPO}} = \min_{p_\theta} -\mathbb{E}_{\mathcal{R}_1^0, \mathcal{R}_2^0 \sim p(\mathcal{R}_1^0 \succ \mathcal{R}_2^0)} p_\theta(\mathcal{R}_1^0 \succ \mathcal{R}_2^0)
$$

$$
= \max_{p_\theta} \mathbb{E}_{\mathcal{R}_1^0, \mathcal{R}_2^0 \sim p(\mathcal{R}_1^0 \succ \mathcal{R}_2^0)} p_\theta(\mathcal{R}_1^0 \succ \mathcal{R}_2^0)
$$

$$
= \min_{p_\theta} \mathbb{D}_{\mathrm{KL}}\left( p(\mathcal{R}_1^0 \succ \mathcal{R}_2^0) \middle\| p_\theta(\mathcal{R}_1^0 \succ \mathcal{R}_2^0) \right)
$$

Again with Gibb's inequality, we can easily identify that $p_{\boldsymbol{\theta}}(\mathcal{R}_1^0 \succ \mathcal{R}_2^0) = p(\mathcal{R}_1^0 \succ \mathcal{R}_2^0)$ achieves the minimum. Thus $p^*(\mathcal{R}^0) = \frac{1}{Z} p_{\text{ref}}(\mathcal{R}^0) \exp\left(\frac{1}{\beta} r(\mathcal{R}^0)\right)$ is also the optimal solution of $L_{\text{DPO}}$.

# D   Implementation Details

## D.1   Model Details

The architecture of the diffusion model used in our method is the same as Luo et al. [36]. The input of the model is the perturbed CDR-H3 and its surrounding context, i.e., 128 nearest residues of the antigen or the antibody framework around the residues of CDR-H3. The input is composed of single residue embeddings and pairwise embeddings. The single residue embedding encodes the information of its amino acid types, torsional angles, and 3D coordinates of all heavy atoms. The pairwise embedding encodes the Euclidean distances and dihedral angles between the two residues. The sizes of the single residue feature and the residue-pair features are 1285 and 64, respectively. Then the features are processed by Multiple Layer Perceptrons (MLPs). The number of layers is 6. The size of the hidden state in the layers is 128. The output of the model is the predicted categorical distribution of amino acid types, $C_\alpha$ coordinates, and a $so(3)$ vector for the rotation matrix.

The number of diffusion steps is 100. We use the cosine $\beta$ schedule with $s = 0.01$ suggested in Ho et al. [20] for amino acid types, $C_\alpha$ coordinates, and orientations.

## D.2   Training Details

**Pre-training**   Following Luo et al. [36], the diffusion model is first trained via the gradient descent method Adam [27] with `init_learning_rate=1e-4`, `betas=(0.9,0.999)`, `batch_size=16`, and `clip_gradient_norm=100`. During the training phase, the weight of rotation loss, position loss, and sequence loss are each set to $1.0$. We also schedule to decay the learning rate multiplied by a factor of $0.8$ and a minimum learning rate of $5e - 6$. The learning rate is decayed if there is no improvement for the validation loss in 10 evaluations. The evaluation is performed for every 1000 training steps. We trained the model on one NVIDIA A100 80G GPU and it could converge within 30 hours and 200k steps.

**Test set**   The original RAbD dataset contains 60 antibody-antigen complexes. In this study, we hope all the complex consists of an antibody heavy chain and a light chain, and at least one protein antigen chain. In practice, **2ghw** and **3uzq** lack light chains, while **3h3b** lacks heavy chains. **5d96** was excluded because of the incorrect chain ID information in rabd_summary.jsonl[3], where heavy chain *J* and light chain *I* do not bind to antigen chain *A*. As for **4etq**, we actually conducted the training (CDR $E_{\text{total}}$=70.55, CDR-Ag $\Delta G$=-4.57), but HERN reported an error when running for this complex, so we did not report it.

**Pair data construction**   In terms of the construction of "winning" and "losing" data pair, we did not pre-define "prefered" and "non-prefered" datasets but rather constructed a unified data pool. During each training step, the paired data used for DPO training is randomly sampled from the data pool. Although their energies and properties have been pre-calculated, the "winning" and "losing" labels are determined in real time. In practice, we used several labels, involving three different preferences related to energy and two preferences related to non-energy-based properties. The "winning" and "losing" labels among these preferences are not necessarily consistent. Therefore, the loss for each type of energy/preference is calculated separately and then aggregated with different weights to update the entire model. Moreover, as the training progresses, we continuously sample new data, calculate their energy, add them to the data pool, and discard some of the older post-added data simultaneously to ensure that the data stays in sync with the policy.

**Fine-tuning**   For ABDPO fine-tuning, the pre-trained diffusion model is further fine-tuned via the gradient descent method Adam with `init_learning_rate=1e-5`, `betas=(0.9,0.999)`, and `clip_gradient_norm=100`. The batch size is 48. More specifically, in a batch, there are 48 pairs of preference data. We do not use a decay learning rate and do not use weight decay in the fine-tuning process. And we use $\beta = 0.01$ and $0.005$ in Eq. (8). We use the hyperparameter search space as follows. As for the three energies introduced in Sec. 4.1, we use 8:8:2 to reweight them (i.e., $\text{Res}_{\text{CDR}}$ $E_{\text{total}}$, $\text{Res}_{\text{CDR}}$-Ag $E_{\text{nonRep}}$, and $\text{Res}_{\text{CDR}}$-Ag $E_{\text{Rep}}$), and reweight pLL and PHR in ABDPO+

---

[3]`https://github.com/THUNLP-MT/MEAN/blob/main/summaries/rabd_summary.jsonl`

to 1. In practice, different antibody-antigen complexes prefer different hyperparameters. For a fair comparison with baselines, we do not carefully picked the optimal hyperparameter for each complex but use a uniform hyperparameter. We fine-tune the pre-trained diffusion model on four NVIDIA A800 40G GPUs for 1,800 steps for each antigen, separately.

### D.3 Ranking Strategy

To rank the numerous generated antibodies with multiple energy labels, we applied a simple ranking strategy based on single energy metrics. The CDR $E_{\text{total}}$ and the CDR-Ag $\Delta G$ of each antibody are ranked independently. Then, a composite ranking score for each antibody is defined as the sum of its CDR $E_{\text{total}}$ rank and CDR-Ag $\Delta G$ rank (for ABDPO+, PHR and pLL are also involved). Finally, the antibodies are ranked according to these composite scores. We acknowledge that this ranking strategy has several limitations. For instance:

1. Equal weights are assigned to all energy types and properties, despite them having differing importance in reality.

2. The distribution patterns of different energy types and properties can vary, with these distributions usually being non-uniform. This could result in scenarios where minor numerical differences in the top-ranking CDR-Ag $\Delta G$ values coincide with larger differences in CDR $E_{\text{total}}$, potentially leading to the selection of samples with suboptimal CDR $E_{\text{total}}$.

However, addressing these issues would require extensive and in-depth exploration of antibody binding mechanisms and energy calculation methodologies. We chose this straightforward, yet impartial, ranking strategy for two key reasons:

1. The primary goal of this work is to reformulate the antibody design task as an energy-focused optimization problem and propose a feasible implementation, rather than to delve into the mechanisms of antibody-antigen binding;

2. Our approach is designed to avoid introducing statistical biases or preferences based on potentially erroneous prior knowledge or favoritism towards particular antibody design methods.

## E    More Evaluation Results

### E.1    Evaluation Results for Ranked Top-1 Design

In Tab. 1, we have reported the average results of all antibodies designed by our method and other baselines. Here we provide the evaluation results for the ranked top-1 design in Tab. 3 (refer to the ranking strategy in Appendix D.3).

Table 3: Average performance of top-1 designs of 55 complexes designed by baselines and our model.

| Methods | CDR $E_{\text{total}}$ ($\downarrow$) | CDR-Ag $\Delta G$ ($\downarrow$) | PHR ($\downarrow$) | pLL ($\uparrow$) | AAR ($\uparrow$) | RMSD ($\downarrow$) |
|---|---|---|---|---|---|---|
| RAbD | 5.25 | -13.04 | 45.78% | -2.20 | 100.00% | 0.00 |
| HERN | 8495.56 | 1296.22 | 48.18% | -2.01 | 33.29% | 9.21 |
| MEAN | 3867.47 | 207.99 | 36.91% | -1.72 | 35.18% | 1.70 |
| dyMEAN | 2987.93 | 1283.97 | 46.27% | -1.79 | **40.74%** | 1.81 |
| DiffAb | 381.82 | 58.84 | 49.19% | -2.03 | 37.99% | 1.62 |
| ABDPO | **68.51** | **-4.96** | 69.97% | -2.15 | 32.92% | **1.58** |
| ABDPO+ | 332.10 | 29.27 | **32.81%** | **-1.54** | 39.55% | 1.67 |

### E.2    Detailed Evaluation Results for each Complex

In Tab. 4 and Tab. 5, we list the CDR $E_{\text{total}}$, CDR-Ag $\Delta G$, PHR and pLL of the reference antibody in RAbD and the average/ranked top-1 antibodies designed by HERN, MEAN, dyMEAN, DiffAb, ABDPO, and ABDPO+ for each complex in the test set separately. In Tab. 5, we highlight the energy values of the designed complexes that surpass the natural one in terms of two energies simultaneously with **bold text**.

Table 4: Detailed evaluation results for reference antibodies and average evaluation results for antibodies designed by HERN, MEAN, dyMEAN, DiffAb, AbDPO, and AbDPO+ for 55 complexes. The data source is the same as that in Tab. 1. For simplicity, we use A, B, C, and D to stand for CDR $E_{total}$, CDR-Ag $\Delta G$, PHR, and pLL respectively in this table. The unit of the two energies is kcal/mol and omitted for brevity.

| PDB id | RAbD (Reference) A | B | C | D | HERN A | B | C | D | MEAN A | B | C | D | dyMEAN A | B | C | D | DiffAb A | B | C | D | AbDPO A | B | C | D | AbDPO+ A | B | C | D |
|---|---|---|---|---|---|---|---|---|---|---|---|---|---|---|---|---|---|---|---|---|---|---|---|---|---|---|---|---|
| 1a14 | 62.28 | -4.72 | 40.00% | -1.56 | 5084.28 | 163.75 | 40.87% | -1.57 | 7614.74 | 280.22 | 31.60% | -1.66 | 5284.77 | 187.78 | 37.71% | -1.93 | 1314.20 | 534.21 | 31.35% | -1.86 | 336.02 | 88.64 | 76.84% | -2.06 | 800.86 | 334.38 | 36.28% | -1.78 |
| 1a2y | -22.18 | -4.81 | 20.00% | -1.30 | 8082.04 | 236.89 | 42.08% | -1.27 | 3722.91 | 75.93 | 35.31% | -1.27 | 70.82 | -0.04 | 60.00% | -1.15 | 538.97 | 50.06 | 37.19% | -1.71 | 247.86 | 9.25 | 48.18% | -1.93 | 428.22 | 50.79 | 26.61% | -1.74 |
| 1fe8 | 25.79 | -14.84 | 44.44% | -2.61 | 4920.00 | 6.17 | 42.48% | -2.15 | 3914.47 | 885.65 | 28.82% | -2.15 | 909.90 | 255.99 | 55.21% | -1.89 | 1239.61 | 600.91 | 45.66% | -1.89 | 880.26 | 257.83 | 48.49% | -2.43 | 1010.34 | 382.26 | 35.99% | -2.31 |
| 1ic7 | 19.96 | -2.62 | 57.14% | -2.89 | 115.08 | 41.72 | 41.77% | -1.84 | 4922.51 | 684.45 | 52.14% | -2.29 | 861.34 | 921.96 | 59.64% | -1.75 | 1277.45 | 1218.19 | 49.58% | -2.36 | 291.91 | 73.22 | 79.27% | -2.15 | 744.88 | 686.06 | 63.91% | -2.24 |
| 1iqd | 43.36 | -5.47 | 70.00% | -2.66 | 10278.46 | 281.72 | 43.07% | -2.29 | 5909.10 | 773.06 | 53.85% | -2.50 | 182.40 | 170.56 | 48.07% | -2.05 | 665.83 | 211.78 | 42.86% | -1.45 | 236.16 | 75.03 | 85.19% | -1.88 | 318.16 | 36.02 | 51.34% | -1.48 |
| 1n8z | 41.88 | -8.00 | 53.85% | -2.50 | 7478.36 | 143.92 | 38.98% | -2.15 | 2623.55 | 28.57% | | -2.15 | 5373.85 | 170.56 | 39.98% | | 1366.55 | 1211.78 | 48.00% | -2.05 | 522.38 | 79.27% | | -2.28 | 983.06 | 591.70 | 38.94% | -2.13 |
| 1ncb | 29.72 | -11.94 | 38.46% | -2.35 | 15891.62 | 1045.70 | 39.66% | -1.94 | 7830.64 | 2181.26 | 19.63% | -1.94 | 7676.52 | 4898.46 | 32.33% | | 2757.11 | 2413.60 | 37.78% | -2.19 | 1226.72 | 597.29 | 68.07% | -2.24 | 2307.18 | 1598.22 | 45.27% | -2.14 |
| 1osp | -1.39 | -15.94 | 42.86% | -1.78 | 22779.04 | 7299.27 | 43.54% | -2.21 | 7299.27 | 2093.95 | 40.33% | -2.21 | 10450.61 | 40.29% | | -2.33 | 1726.43 | 1163.36 | 38.39% | -2.33 | 741.85 | 281.07 | 64.96% | -2.27 | 1354.85 | 393.58 | 35.34% | -2.13 |
| 1uj3 | -12.93 | -11.45 | 40.00% | -2.16 | 12931.58 | 24.87 | 42.86% | -2.22 | 3651.92 | 833.83 | 26.09% | | 725.91 | 60.68 | 59.69% | -1.46 | 1201.06 | 515.24 | 46.46% | -2.25 | 548.70 | 309.96 | 60.94% | -2.50 | 902.98 | 535.91 | 40.94% | -2.12 |
| 1w72 | 8.36 | -16.06 | 46.67% | -2.05 | 13064.14 | 236.14 | 38.19% | -1.97 | 9270.47 | 359.62 | 29.31% | -1.97 | 4646.75 | 1691.72 | 32.78% | -1.66 | 1898.70 | 2301.58 | 46.77% | -2.10 | 541.99 | 323.45 | 59.17% | -1.86 | 1475.37 | 1319.43 | 45.63% | -1.82 |
| 2adf | -20.47 | -15.53 | 36.36% | -2.16 | 9963.24 | 668.11 | 41.43% | -2.11 | 5242.10 | 243.62 | 30.30% | -1.90 | 2177.12 | 737.82 | 51.33% | -1.79 | 1572.54 | 348.72 | 44.23% | -2.16 | 1211.16 | 1125.38 | 49.15% | -2.42 | 1863.22 | 1872.10 | 31.91% | -2.32 |
| 2b2x | 5.41 | -0.90 | 58.33% | -2.22 | 10070.26 | 1046.74 | 41.10% | -2.16 | 10872.33 | 92.94 | 18.79% | -2.16 | 2568.95 | 1049.25 | 50.00% | -1.78 | 921.64 | 114.18 | 50.00% | -2.15 | 627.10 | 120.47 | 76.78% | -2.15 | 1033.04 | 272.63 | 47.01% | -1.93 |
| 2cmr | 5.25 | -9.79 | 41.67% | -2.20 | 15455.06 | 1146.95 | 42.49% | -2.24 | 6012.22 | 1194.67 | 26.30% | -1.82 | 3176.59 | 1084.35 | 49.39% | -1.78 | 2254.27 | 1702.79 | 49.39% | -2.18 | 1493.98 | 908.39 | 54.64% | -1.96 | 1670.16 | 1134.01 | 42.45% | -1.84 |
| 2dd8 | 68.27 | -7.21 | 63.64% | -2.27 | 10822.48 | 1265.56 | 41.95% | -2.14 | 6360.61 | 261.43 | 22.25% | -2.14 | 1868.70 | 230.60 | 53.98% | -1.62 | 921.64 | 1211.78 | 50.00% | -1.99 | 304.55 | 11.63 | 63.50% | -2.11 | 435.60 | 49.50 | 54.50% | -2.04 |
| 2vxt | -10.32 | -12.95 | 66.67% | -1.76 | 5017.31 | 112.94 | 46.18% | -2.20 | 1378.61 | 198.35 | 16.93% | -1.11 | 230.60 | 170.56 | 50.00% | -1.11 | 1286.01 | 523.04 | 51.22% | -2.11 | 378.06 | 127.33 | 87.76% | -2.11 | 523.04 | 303.94 | 67.54% | -1.62 |
| 2xqy | -3.67 | -16.14 | 54.55% | -2.68 | 11783.79 | 1547.70 | 41.34% | -2.07 | 4532.76 | 633.59 | 35.23% | -1.38 | 1831.64 | 1248.24 | 45.45% | -2.19 | 975.16 | 576.50 | 35.32% | -2.18 | 521.87 | 171.23 | 57.48% | -2.12 | 1093.36 | 508.48 | 35.79% | -2.17 |
| 2xwt | -19.96 | -27.99 | 50.00% | -2.57 | 14800.89 | 1153.82 | 38.63% | -2.11 | 6877.42 | 3150.51 | 25.65% | -2.11 | 4033.71 | 4267.90 | 42.66% | -2.19 | 1941.55 | 1394.53 | 42.19% | -2.11 | 279.27 | 40.72 | 80.77% | -2.13 | 900.33 | 356.91 | 52.69% | -1.94 |
| 2ypv | 4.72 | -6.94 | 25.00% | -1.43 | 17470.94 | 904.99 | 40.49% | -2.12 | 5817.70 | 2291.43 | 27.82% | -1.48 | 5638.27 | 6442.33 | 30.99% | -1.25 | 2312.54 | 2615.65 | 37.98% | -2.13 | 409.78 | 151.56 | 83.33% | -2.24 | 1161.97 | 598.35 | 49.83% | -2.06 |
| 3bn9 | 81.92 | -0.89 | 33.33% | -1.71 | 12219.73 | 542.93 | 43.23% | -2.24 | 4956.22 | 92.90 | 26.50% | -2.24 | 8516.00 | 0.20 | 66.67% | -1.91 | 1911.65 | 261.66 | 49.13% | -1.91 | 717.15 | 46.78 | 77.90% | -1.81 | 1086.68 | 172.95 | 38.04% | -2.30 |
| 3cx5 | -18.25 | -14.91 | 53.85% | -1.80 | 18070.35 | 1402.27 | 39.48% | -1.46 | 6987.09 | 303.81 | 38.54% | -1.46 | 5257.45 | 62.91 | 35.45% | -1.26 | 1652.10 | 421.00 | 38.65% | -1.84 | 388.32 | 89.93 | 74.93% | -1.81 | 923.14 | 235.49 | 44.65% | -1.80 |
| 3ffd | 43.13 | -12.63 | 36.36% | -2.39 | 3076.25 | 542.93 | 42.19% | -2.05 | 2685.49 | 527.96 | 40.48% | -2.05 | 576.62 | 741.54 | 54.55% | -1.76 | 1306.04 | 1877.71 | 42.80% | -2.06 | 621.94 | 694.12 | 46.64% | -2.30 | 872.12 | 1078.66 | 33.99% | -2.11 |
| 3hi6 | -1.47 | -12.35 | 46.15% | -1.93 | 13361.85 | 28.71 | 39.62% | -1.99 | 11746.52 | 6383.11 | 22.16% | -1.99 | 3042.84 | 2664.46 | 47.36% | -2.03 | 2603.22 | 3406.63 | 37.36% | -2.14 | 1347.67 | 1059.13 | 65.30% | -2.07 | 1722.26 | 1465.23 | 43.35% | -1.91 |
| 3k2u | 18.71 | -14.57 | 72.73% | -3.02 | 11409.01 | 28.71 | 41.38% | -2.09 | 6503.22 | 2403.08 | 24.86% | -2.09 | 988.56 | 1328.85 | 54.55% | -1.33 | 1034.26 | 1211.08 | 39.91% | -1.33 | 417.15 | 228.56 | 57.20% | -2.05 | 830.86 | 595.22 | 33.55% | -1.97 |
| 3l95 | -1.18 | -18.48 | 58.33% | -1.63 | 15605.61 | 371.11 | 41.02% | -1.49 | 6733.72 | 1246.90 | 24.83% | -1.49 | 1090.77 | 767.77 | 40.84% | -1.49 | 2869.13 | 40.63 | 40.84% | -2.00 | 447.80 | 95.36 | 70.62% | -1.90 | 817.12 | 251.49 | 51.82% | -1.83 |
| 3mxw | -7.55 | -19.04 | 41.67% | -2.10 | 20920.65 | 726.31 | 37.63% | -1.90 | 6335.74 | 805.41 | 31.90% | -1.90 | 4070.17 | 1968.18 | 33.33% | -1.90 | 1610.09 | 1589.86 | 39.11% | -2.09 | 251.53 | 157.53 | 76.52% | -2.44 | 652.83 | 664.27 | 48.61% | -2.08 |
| 3nid | -21.55 | -28.54 | 41.67% | -2.06 | 11025.32 | 702.13 | 42.75% | -2.08 | 9531.63 | 3817.14 | 21.96% | -2.18 | 1542.49 | 1474.66 | 55.16% | -1.89 | 2327.91 | 3134.96 | 34.81% | -2.14 | 1567.97 | 1795.32 | 55.56% | -2.29 | 2246.97 | 2987.74 | 34.16% | -2.15 |
| 3o2d | 0.23 | -13.42 | 66.67% | -2.01 | 14375.34 | 1747.32 | 39.24% | -1.82 | 5817.70 | 231.91 | 21.96% | -1.86 | 3792.46 | 238.21 | 36.74% | -1.64 | 1968.51 | 671.50 | 37.50% | -2.38 | 590.40 | 52.58 | 77.71% | -2.01 | 1270.30 | 270.11 | 46.28% | -2.01 |
| 3rkd | -6.61 | -10.35 | 43.75% | -1.94 | 3822.72 | 419.59 | 37.37% | -2.11 | 5400.31 | 177.87 | 53.58% | -1.98 | 2224.89 | 28.81 | 37.63% | -2.18 | 2545.63 | 1419.63 | 38.77% | -2.15 | 388.75 | 39.36 | 69.63% | -2.57 | 1140.76 | 106.51 | 34.64% | -2.09 |
| 3s35 | -4.63 | -5.60 | 45.45% | -2.23 | 16862.12 | 410.94 | 44.32% | -2.03 | 3690.15 | 903.86 | 23.33% | -2.03 | 1052.31 | 1200.26 | 57.19% | -1.99 | 1228.31 | 1383.63 | 38.70% | -1.99 | 185.90 | 86.67 | 77.55% | -2.14 | 244.30 | 64.01 | 64.01% | -1.92 |
| 3w9e | -9.93 | -18.41 | 40.00% | -2.29 | 18322.87 | 2687.04 | 39.72% | -1.95 | 9415.71 | 2837.23 | 23.68% | -1.98 | 9644.55 | 3212.83 | 40.00% | -2.16 | 1768.13 | 1320.91 | 45.38% | -2.18 | 1266.82 | 426.19 | 57.95% | -1.97 | 1807.16 | 814.79 | 35.35% | -1.97 |
| 4cmh | -19.18 | -16.54 | 30.77% | -1.63 | 9079.72 | 249.56 | 44.23% | -2.14 | 11848.30 | 1885.59 | 38.86% | -1.95 | 5226.83 | 1468.30 | 53.89% | -1.95 | 2710.24 | 2869.13 | 41.79% | -2.00 | 646.94 | 637.58 | 82.17% | -1.97 | 1976.46 | 235.91 | 45.95% | -1.88 |
| 4dtg | 7.56 | -5.43 | 50.00% | -2.31 | 12267.42 | 667.19 | 38.91% | -1.90 | 5047.79 | 1.41 | 52.72% | -1.90 | 1969.31 | 965.74 | 44.68% | -1.88 | 1753.12 | 1906.32 | 37.98% | -2.09 | 195.38 | 67.03 | 87.13% | -2.09 | 913.31 | 826.23 | 45.76% | -2.02 |
| 4dvr | -6.74 | 1.13 | 66.67% | -2.89 | 11025.32 | 16.41 | 39.63% | -2.10 | 4932.19 | 89.35 | 39.63% | -2.10 | 3080.76 | 972.57 | 33.46% | -2.57 | 860.12 | 511.29 | 38.50% | -2.14 | 212.96 | 78.48 | 67.27% | -2.47 | 474.54 | 235.55 | 40.15% | -2.25 |
| 4ffv | 28.69 | 0.67 | 50.00% | -2.96 | 3822.72 | 164.84 | 43.91% | -1.96 | 2064.08 | 53.98 | 20.89% | -1.96 | 517.77 | -0.47 | 69.95% | -1.48 | 712.45 | 71.58 | 38.96% | -1.87 | 247.26 | 0.49 | 73.39% | -1.78 | 462.92 | 15.58 | 53.49% | -1.69 |
| 4iqj | 33.50 | -21.93 | 38.89% | -1.66 | 16862.12 | 308.56 | 40.08% | -2.08 | 6404.03 | 1009.16 | 38.77% | -2.08 | 7140.47 | 993.46 | 41.23% | -2.04 | 3534.93 | 3314.58 | 40.10% | -2.15 | 1675.57 | 703.82 | 70.95% | -2.36 | 2682.46 | 1371.78 | 36.75% | -2.00 |
| 4g6j | 0.30 | -8.81 | 45.45% | -1.92 | 11113.87 | 720.86 | 42.42% | -2.23 | 9305.46 | 586.48 | 28.97% | -2.06 | 951.58 | 875.88 | 62.78% | -1.28 | 1968.51 | 658.75 | 40.93% | -1.97 | 375.88 | 67.67 | 81.87% | -1.97 | 1239.99 | 178.80 | 70.03% | -2.12 |
| 4h8w | -8.60 | -21.61 | 50.00% | -2.64 | 11453.29 | 155.15 | 38.06% | -2.00 | 5037.48 | 813.40 | 30.38% | -2.00 | 2035.59 | 1358.96 | 47.35% | -2.01 | 848.30 | 664.43 | 40.19% | -2.04 | 280.47 | 88.75 | 66.93% | -2.05 | 350.47 | 297.03 | 47.83% | -1.96 |
| 4ki5 | -1.33 | -12.71 | 50.00% | -1.84 | 9596.77 | 229.00 | 43.81% | -2.28 | 2602.18 | 382.40 | 29.13% | -2.05 | 304.67 | 89.91 | 46.99% | -1.72 | 838.21 | 268.25 | 28.78% | -2.01 | 618.75 | 171.74 | 60.98% | -2.00 | 264.67 | 235.46 | 42.45% | -2.00 |
| 4lvn | -8.15 | -16.58 | 26.67% | -1.80 | 13321.21 | 2276.21 | 39.46% | -1.95 | 8141.08 | 69.59 | 39.48% | -1.93 | 4663.15 | -0.12 | 33.85% | -1.51 | 2494.06 | 2181.34 | 36.35% | -2.12 | 776.62 | 429.83 | 68.58% | -2.42 | 1646.62 | 893.89 | 35.31% | -2.20 |
| 4ot1 | 40.37 | -11.59 | 41.67% | -2.50 | 7876.98 | 741.87 | 37.67% | -2.06 | 6142.84 | 120.48 | 53.86% | -1.98 | 3396.91 | 103.42 | 53.89% | -1.68 | 3260.43 | 511.25 | 46.31% | -1.37 | 1042.90 | 19.44 | 73.28% | -2.20 | 1831.02 | 357.13 | 47.60% | -1.97 |
| 4qci | -11.19 | -25.77 | 41.67% | -2.61 | 9108.96 | 1733.31 | 38.04% | -2.19 | 26107.87 | 2167.05 | 38.04% | -2.19 | 18833.98 | 5614.24 | 39.15% | -1.52 | 3532.15 | 3014.32 | 32.81% | -1.52 | 1384.44 | 1122.83 | 70.01% | -2.27 | 1972.81 | 1404.79 | 38.15% | -2.00 |
| 4xnq | 14.31 | -3.35 | 53.85% | -2.61 | 2648.94 | 841.19 | 39.06% | -2.09 | 5365.28 | 0.06 | 38.97% | -2.09 | 2043.48 | 48.25 | 24.64% | -1.63 | 537.96 | 0.39 | 40.56% | -1.80 | 129.25 | -0.12 | 62.42% | -2.28 | 170.91 | 0.63 | 44.31% | -2.15 |
| 4ydk | -18.37 | -20.88 | 75.00% | -1.89 | 14375.34 | 1055.40 | 40.14% | -2.06 | 6211.06 | 586.48 | 27.67% | -2.06 | 6710.95 | 215.05 | 24.45% | -2.19 | 6710.95 | 541.82 | 34.51% | -2.64 | 582.21 | 125.31 | 74.48% | -2.64 | 274.35 | 274.35 | 34.38% | -2.25 |
| 5b8c | -30.59 | -35.64 | 36.36% | -2.55 | 20568.19 | 203.12 | 38.42% | -2.20 | 25670.41 | 1852.50 | 27.23% | -2.27 | 10877.30 | 2840.50 | 40.41% | -2.20 | 4397.02 | 2551.16 | 31.37% | -2.33 | 1738.58 | 875.15 | 63.92% | -2.31 | 2721.85 | 1232.94 | 33.43% | -2.18 |
| 5bv7 | -4.19 | -15.23 | 38.46% | -1.80 | 5093.61 | 967.04 | 40.42% | -1.45 | 6173.73 | 1110.47 | 29.13% | -1.62 | 4293.39 | 261.41 | 30.77% | -1.50 | 968.21 | 658.75 | 48.87% | -1.62 | 178.36 | 19.48 | 77.04% | -2.28 | 178.80 | 255.05 | 52.84% | -1.70 |
| 5d93 | -15.23 | -8.81 | 44.44% | -1.73 | 11453.29 | 1659.55 | 38.62% | -1.88 | 6485.45 | 382.40 | 56.09% | -1.88 | 4103.66 | 142.34 | 47.29% | -2.12 | 2561.50 | 2520.19 | 36.27% | -2.29 | 1389.85 | 1377.79 | 61.02% | -2.00 | 1727.75 | 1862.98 | 37.53% | -1.58 |
| 5en2 | 44.87 | -8.44 | 35.29% | -1.95 | 9596.77 | 229.00 | 43.81% | -2.28 | 2602.18 | 268.25 | 33.15% | -2.04 | 4739.77 | 89.91 | 32.41% | -2.02 | 2665.76 | 1961.80 | 26.65% | -2.13 | 436.10 | 160.37 | 74.27% | -2.15 | 1236.57 | 184.26 | 35.24% | -1.81 |
| 5f9o | -6.28 | -6.83 | 46.67% | -2.28 | 6967.16 | 741.87 | 37.67% | -2.06 | 6681.43 | 936.62 | 43.81% | -2.06 | 4739.77 | 994.38 | 43.81% | -1.99 | 2028.59 | 1955.59 | 41.70% | -2.13 | 362.41 | 40.84 | 82.55% | -2.43 | 1083.65 | 431.15 | 63.78% | -1.95 |
| 5ggs | -20.91 | -16.16 | 38.46% | -2.07 | 7470.50 | 731.35 | 38.54% | -2.51 | 6708.07 | 992.69 | 28.21% | -2.03 | 5657.30 | 33.28 | 27.19% | -2.32 | 1733.31 | 834.52 | 43.79% | -2.17 | 205.83 | 10.38 | 89.51% | -2.17 | 480.12 | 235.27 | 58.93% | -2.04 |
| 5hi4 | 5.96 | -16.16 | 46.67% | -2.07 | 11977.15 | 731.35 | 42.90% | -2.34 | 5135.00 | 547.59 | 18.84% | -2.03 | 4005.27 | 1021.12 | 29.48% | -1.47 | 1175.68 | 1053.91 | 29.48% | -1.64 | 460.46 | 100.91 | 64.11% | -2.05 | 854.79 | 346.15 | 42.00% | -2.06 |
| 5j13 | -15.18 | -17.75 | 38.46% | -2.17 | 12601.91 | 3531.94 | 37.15% | -1.96 | 6211.06 | 547.59 | 34.72% | -1.90 | 7589.29 | 1116.42 | 34.83% | -2.39 | 1472.07 | 2245.62 | 44.97% | -2.39 | 384.44 | 66.18 | 86.01% | -2.03 | 760.26 | 219.79 | 63.78% | -2.08 |
| 5l6y | 0.48 | -18.76 | 60.00% | -2.34 | 17597.70 | 609.49 | 35.69% | -1.94 | 7367.48 | 349.13 | 17.22% | -1.96 | 4201.51 | 1432.54 | 32.36% | -2.31 | 1322.21 | 384.84 | 45.79% | -2.31 | 557.41 | 26.68 | 66.15% | -2.18 | 920.33 | 79.18 | 42.33% | -1.98 |
| 5mes | 0.61 | -7.42 | 66.67% | -2.39 | 9098.84 | 323.58 | 41.23% | -2.19 | 5031.27 | 89.94 | 31.51% | -2.14 | 1979.26 | 1613.61 | 33.72% | -2.45 | 766.11 | 359.95 | 47.44% | -2.14 | 185.66 | 62.14 | 71.14% | -2.19 | 323.58 | 86.86 | 50.61% | -2.07 |
| 5nuz | -16.73 | -18.19 | 23.08% | -2.16 | 13795.67 | 641.68 | 40.83% | -1.99 | 6690.16 | 1012.48 | 25.64% | -1.99 | 1746.62 | 368.08 | 38.66% | -2.17 | 2199.90 | 2384.32 | 33.21% | -2.14 | 1247.62 | 1057.81 | 54.85% | -2.03 | 1649.31 | 1431.98 | 37.70% | -2.02 |

Table 5: Detailed evaluation results for reference antibodies and top-1 antibodies designed by HERN, MEAN, dyMEAN, DiffAb, AbDPO, and AbDPO+ for 55 complexes. The data source is the same as that in Tab. 3. For simplicity, we use A, B, C, and D to stand for CDR $E_{total}$, CDR-Ag $\Delta G$, PHR, and pLL respectively in this table. The unit of the two energies is kcal/mol and omitted for brevity.

| PDB id | RAbD (Reference) | | | | HERN | | | | MEAN | | | | dyMEAN | | | | DiffAb | | | | AbDPO | | | | AbDPO+ | | | |
| --- | --- | --- | --- | --- | --- | --- | --- | --- | --- | --- | --- | --- | --- | --- | --- | --- | --- | --- | --- | --- | --- | --- | --- | --- | --- | --- | --- | --- |
| | A | B | C | D | A | B | C | D | A | B | C | D | A | B | C | D | A | B | C | D | A | B | C | D | A | B | C | D |
| 1a14 | 62.28 | -4.72 | 40.00% | -1.56 | 3370.02 | -1.79 | 26.67% | -1.01 | 5142.76 | -2.97 | 26.67% | -1.81 | 3521.78 | -2.04 | 46.67% | -1.80 | 298.23 | -6.50 | 20.00% | -1.97 | 44.65 | -5.83 | 86.67% | -2.25 | 189.19 | 1.23 | 26.67% | -1.02 |
| 1a2y | -22.18 | -4.81 | 20.00% | -1.30 | 6101.92 | -1.35 | 50.00% | -1.46 | 2259.30 | -1.43 | 40.00% | -1.46 | 61.49 | -0.06 | 60.00% | -1.15 | 125.47 | -1.84 | 40.00% | -1.12 | 20.87 | -3.58 | 60.00% | -2.43 | 74.95 | -5.61 | 20.00% | -1.35 |
| 1fe8 | 25.79 | -14.84 | 44.44% | -2.61 | 3663.52 | -1.34 | 55.56% | -2.16 | 1721.37 | 3.69 | 44.44% | -1.69 | 823.80 | 221.87 | 55.56% | -1.93 | 332.67 | 14.57 | 44.44% | -2.03 | 127.38 | -8.11 | 33.33% | -2.49 | 667.06 | -6.12 | 22.22% | -1.75 |
| 1ic7 | 19.96 | -2.62 | 57.14% | -2.89 | 4871.00 | -2.07 | 42.86% | -2.08 | 525.84 | -2.26 | 28.57% | -2.03 | 413.21 | 60.26 | 42.86% | -1.45 | 131.09 | -1.73 | 42.86% | -2.15 | -1.63 | -3.50 | 71.43% | -1.43 | 8.08 | -3.57 | 42.86% | -0.89 |
| 1iqd | 43.36 | -5.47 | 70.00% | -2.66 | 8950.62 | -0.63 | 50.00% | -1.99 | 1343.07 | -3.17 | 70.00% | -3.13 | 502.21 | -1.48 | 60.00% | -1.97 | 229.08 | -1.67 | 50.00% | -2.06 | 12.71 | -7.36 | 90.00% | -2.22 | 131.10 | 47.91 | 50.00% | -1.81 |
| 1n8z | 41.88 | -8.00 | 53.85% | -2.50 | 5547.03 | 6.55 | 66.67% | -2.08 | 2886.71 | 5.78 | 23.08% | -2.34 | 1804.53 | -2.52 | 23.08% | -2.05 | 326.22 | -0.53 | 38.46% | -2.18 | 159.57 | -3.20 | 38.46% | -2.38 | 101.78 | 40.02 | 30.77% | -1.51 |
| 1ncb | 29.72 | -11.94 | 38.46% | -2.35 | 13605.15 | 357.59 | 46.15% | -1.95 | 4130.61 | 624.22 | 23.08% | -1.27 | 6597.50 | 4301.46 | 30.77% | -2.15 | 631.94 | 150.24 | 53.85% | -2.25 | 59.01 | -6.33 | 76.92% | -2.19 | 1095.23 | 2.21 | 30.77% | -1.81 |
| 1osp | -1.39 | -15.94 | 42.86% | -1.78 | 17731.35 | 32650.25 | 57.14% | -1.74 | 2693.69 | 34.23 | 57.14% | -1.88 | 8881.64 | 12416.92 | 50.00% | -1.88 | 237.19 | 9.82 | 57.14% | -2.49 | 34.84 | -5.55 | 78.57% | -2.18 | 184.27 | -4.81 | 28.57% | -1.37 |
| 1uj3 | -12.93 | -11.45 | 40.00% | -2.16 | 11290.58 | -0.51 | 60.00% | -2.05 | 1709.16 | -2.04 | 50.00% | -1.88 | 660.71 | 36.96 | 60.00% | -1.68 | 264.93 | 16.69 | 40.00% | -2.28 | 119.22 | -5.71 | 60.00% | -2.42 | 216.60 | 0.72 | 20.00% | -1.91 |
| 1w72 | -8.36 | -16.06 | 46.67% | -2.05 | 10076.18 | 88.75 | 40.00% | -1.92 | 4196.89 | 0.74 | 26.67% | -1.49 | 4162.03 | 1206.78 | 33.33% | -2.34 | 386.12 | 4.72 | 66.67% | -1.95 | 82.61 | -1.51 | 80.00% | -1.88 | 739.00 | 165.90 | 33.33% | -1.27 |
| 2adf | -20.47 | -15.53 | 36.36% | -2.22 | 8781.28 | -2.78 | 45.45% | -2.16 | 2996.23 | 361.67 | 27.27% | -1.89 | 788.08 | 81.23 | 54.55% | -1.07 | 174.54 | 9.10 | 54.55% | -2.11 | 83.46 | -4.01 | 63.64% | -2.29 | 787.24 | -3.84 | 41.67% | -1.87 |
| 2b2x | 5.41 | -0.90 | 58.33% | -2.16 | 7922.20 | -1.74 | 50.00% | -2.44 | 7299.02 | -3.12 | 25.00% | -1.54 | 2916.38 | 2284.43 | 41.67% | -1.54 | 479.11 | -1.18 | 66.67% | -2.09 | 134.20 | -4.00 | 66.67% | -2.26 | 257.80 | -4.26 | 36.36% | -1.57 |
| 2cmr | 5.25 | -9.79 | 41.67% | -2.20 | 13987.00 | 88.57 | 50.00% | -2.00 | 3548.46 | 201.80 | 33.33% | -1.18 | 3024.63 | 869.16 | 50.00% | -1.78 | 341.10 | 17.00 | 66.67% | -2.53 | 214.55 | -6.55 | 50.00% | -1.74 | 502.95 | 16.40 | 33.33% | -1.62 |
| 2dd8 | 68.27 | -7.21 | 63.64% | -2.27 | 8801.59 | 131.33 | 72.73% | -2.43 | 4785.61 | -5.17 | 18.18% | -1.59 | 1814.99 | 960.05 | 54.55% | -1.11 | 214.55 | -4.85 | 36.36% | -2.25 | 28.07 | -8.74 | 63.64% | -2.21 | 343.40 | -3.19 | 36.36% | -1.62 |
| 2vxt | -3.67 | -12.95 | 66.67% | -1.76 | 4792.96 | -0.78 | 66.67% | -2.53 | 645.86 | 14.70 | 66.67% | -1.72 | 191.22 | 117.11 | 66.67% | -1.11 | 48.87 | -2.00 | 66.67% | -1.61 | 2.26 | -7.45 | 63.64% | -2.11 | 16.92 | -4.38 | 33.33% | -1.35 |
| 2xqy | 4.72 | -16.14 | 54.55% | -2.68 | 9737.32 | -0.83 | 36.36% | -1.94 | 1266.63 | -0.72 | 63.64% | -1.63 | 1528.26 | 550.52 | 45.45% | -0.76 | 222.91 | -1.87 | 36.36% | -2.40 | 23.11 | -4.60 | 66.67% | -2.11 | 63.48 | -3.19 | 36.36% | -1.34 |
| 2xwt | -19.96 | -27.99 | 50.00% | -2.57 | 11584.12 | -0.06 | 50.00% | -2.14 | 4082.86 | 104.45 | 16.67% | -1.94 | 3180.46 | 2638.26 | 33.33% | -2.36 | 485.02 | 4.74 | 41.67% | -1.87 | 35.10 | -5.65 | 84.62% | -1.94 | 349.68 | -1.46 | 33.33% | -1.63 |
| 2ypv | 81.92 | -6.94 | 25.00% | -1.43 | 15216.10 | 86.25 | 25.00% | -1.77 | 3475.66 | -2.19 | 33.33% | -1.41 | 4417.05 | 6151.36 | 25.00% | -1.10 | 621.23 | 295.98 | 41.67% | -1.94 | 65.47 | -2.90 | 66.67% | -2.28 | 227.20 | 119.80 | 33.33% | -1.63 |
| 3bn9 | -18.25 | -0.89 | 33.33% | -1.71 | 11004.75 | 0.37 | 55.56% | -2.06 | 3153.02 | -2.19 | 44.44% | -1.94 | 7742.77 | -0.36 | 66.67% | -1.91 | 573.75 | -2.38 | 55.56% | -2.04 | 59.96 | -3.11 | 55.56% | -1.36 | 362.72 | -1.73 | 22.22% | -1.49 |
| 3cx5 | -18.25 | -14.91 | 33.33% | -1.80 | 14437.58 | 462.31 | 40.00% | -1.67 | 4322.29 | -5.07 | 46.67% | -1.22 | 4842.11 | 45.60 | 40.00% | -1.37 | 134.10 | -4.86 | 46.67% | -1.61 | 35.44 | -10.11 | 45.45% | -2.32 | 405.94 | -5.34 | 26.67% | -1.07 |
| 3ffd | 43.13 | -12.63 | 36.36% | -2.39 | 2076.23 | -3.98 | 45.45% | -2.24 | 1419.27 | 1.47 | 54.55% | -1.75 | 502.53 | 615.53 | 54.55% | -1.76 | 125.43 | 164.91 | 45.45% | -1.98 | 40.23 | -1.87 | 45.45% | -2.32 | 175.01 | 81.20 | 18.18% | -1.79 |
| 3hi6 | -1.47 | -12.35 | 46.15% | -1.93 | 11018.30 | -6.46 | 53.85% | -1.92 | 6648.87 | 2889.20 | 23.08% | -1.92 | 2511.50 | 1888.22 | 61.54% | -1.73 | 536.27 | 243.63 | 61.54% | -2.07 | 149.54 | -5.65 | 84.62% | -1.87 | 388.65 | 81.17 | 38.46% | -1.73 |
| 3k2u | 18.71 | -14.57 | 72.73% | -3.02 | 8174.07 | -1.96 | 54.55% | -2.23 | 3213.40 | 1028.78 | 36.36% | -2.23 | 886.48 | 1171.60 | 54.55% | -1.00 | 136.63 | 35.80 | 45.45% | -1.74 | 18.86 | -3.10 | 63.64% | -2.10 | 201.56 | 8.50 | 27.27% | -1.62 |
| 3l95 | -1.18 | -18.48 | 58.33% | -2.50 | 13351.29 | -1.70 | 41.67% | -2.15 | 4472.76 | 115.65 | 41.67% | -2.15 | 968.84 | 276.55 | 66.67% | -1.98 | 312.03 | 1.23 | 66.67% | -1.70 | 73.09 | -7.27 | 58.33% | -1.56 | 206.67 | -1.48 | 33.33% | -1.35 |
| 3mxw | -7.55 | -19.04 | 41.67% | -2.10 | 6712.93 | 7.23 | 33.33% | -1.92 | 3141.84 | 4.10 | 33.33% | -1.92 | 3247.51 | 1354.59 | 33.33% | -2.27 | 172.90 | 4.59 | 66.67% | -2.00 | 29.67 | -4.00 | 83.33% | -2.39 | 94.21 | -2.64 | 41.67% | -1.61 |
| 3nid | -21.55 | -28.54 | 41.67% | -2.06 | 8480.65 | 123.17 | 50.00% | -2.22 | 6265.25 | 1813.79 | 25.00% | -2.22 | 685.91 | 626.38 | 58.33% | -1.82 | 540.87 | 407.11 | 50.00% | -1.92 | 59.61 | -4.63 | 75.00% | -2.56 | 926.30 | 467.72 | 25.00% | -1.85 |
| 3o2d | 0.23 | -13.42 | 46.67% | -2.01 | 4273.90 | 735.38 | 46.67% | -1.77 | 4629.39 | -1.23 | 40.00% | -1.77 | 2530.49 | -0.98 | 46.67% | -1.53 | 388.39 | 0.90 | 26.67% | -2.40 | 49.28 | -3.62 | 93.33% | -2.33 | 235.27 | -1.20 | 40.00% | -1.28 |
| 3rkd | -6.61 | -10.35 | 43.75% | -1.94 | 1818.15 | 124.59 | 37.50% | -2.07 | 4126.58 | -4.75 | 56.25% | -2.07 | 1748.64 | 6.24 | 37.50% | -2.18 | 576.71 | 7.22 | 43.75% | -2.00 | 77.66 | -5.85 | 75.00% | -2.71 | 337.99 | -3.64 | 25.00% | -1.68 |
| 3s35 | -4.63 | -5.60 | 20.00% | -2.23 | 6506.10 | 7.92 | 60.00% | -2.22 | 1638.19 | 99.18 | 30.00% | -2.22 | 919.12 | 962.83 | 60.00% | -1.44 | 75.14 | 22.60 | 60.00% | -1.67 | 11.55 | -7.60 | 80.00% | -2.16 | 36.60 | -4.65 | 60.00% | -1.69 |
| 3w9e | -9.93 | -18.41 | 40.00% | -2.29 | 14363.44 | 799.14 | 46.67% | -1.98 | 4551.50 | 518.30 | 33.33% | -1.98 | 8590.08 | 11529.24 | 40.00% | -2.17 | 545.30 | 26.61 | 33.33% | -1.76 | 103.21 | -5.46 | 73.33% | -1.89 | 555.46 | 3.26 | 33.33% | -1.08 |
| 4cmh | -19.18 | -16.54 | 30.77% | -1.63 | 7821.21 | 60.38 | 46.15% | -1.81 | 6314.53 | 424.01 | 38.46% | -1.81 | 4680.21 | 1388.22 | 30.77% | -1.95 | 710.43 | 774.04 | 53.85% | -1.97 | 50.34 | 1.41 | 84.62% | -1.34 | 671.25 | 145.64 | 46.15% | -1.34 |
| 4dtg | 7.56 | -5.43 | 50.00% | -2.31 | 9843.25 | 16234.71 | 50.00% | -2.01 | 1817.18 | -0.84 | 57.14% | -2.01 | 1259.81 | 73.30 | 50.00% | -1.84 | 77.78 | 20.77 | 35.71% | -2.12 | 41.75 | -4.41 | 85.71% | -2.06 | 127.02 | 3.63 | 35.71% | -1.76 |
| 4dvr | -6.74 | 1.13 | 66.67% | -2.89 | 8851.78 | -2.11 | 66.67% | -1.74 | 2228.20 | -1.62 | 41.67% | -1.21 | 2315.58 | 790.43 | 33.33% | -2.57 | 189.44 | -2.10 | 50.00% | -1.95 | 13.89 | -5.43 | 75.00% | -2.51 | 70.08 | -1.22 | 33.33% | -1.84 |
| 4ffv | 28.69 | 0.67 | 50.00% | -2.96 | 1142.48 | -0.89 | 50.00% | -2.25 | 1265.17 | -4.00 | 20.00% | -1.63 | 425.73 | -0.51 | 70.00% | -1.48 | 151.06 | -1.29 | 50.00% | -1.71 | 44.35 | -2.34 | 60.00% | -2.19 | 96.61 | -3.81 | 50.00% | -1.51 |
| 4fqi | 33.50 | -21.93 | 38.89% | -1.66 | 12660.96 | 48.19 | 44.44% | -1.94 | 4163.98 | 87.78 | 38.89% | -1.94 | 5274.34 | 531.42 | 44.44% | -2.14 | 966.00 | 475.91 | 38.89% | -1.81 | 151.09 | 1.40 | 77.78% | -2.13 | 869.39 | 267.80 | 16.67% | -1.68 |
| 4g6j | -4.19 | -8.81 | 45.45% | -1.80 | 8685.25 | 34.14 | 45.45% | -2.13 | 3910.08 | 189.16 | 18.18% | -2.13 | 796.48 | 628.64 | 63.64% | -1.29 | 212.73 | 0.48 | 63.64% | -2.02 | 31.88 | -3.53 | 81.82% | -2.03 | 72.92 | -3.63 | 54.55% | -1.62 |
| 4g6m | -8.60 | -21.61 | 50.00% | -2.64 | 5173.21 | -0.57 | 41.67% | -2.04 | 2615.09 | 74.92 | 41.67% | -2.04 | 1650.47 | 1053.70 | 50.00% | -2.09 | 14.30 | -1.55 | 41.67% | -2.01 | 11.82 | -5.27 | 66.67% | -1.90 | 20.32 | -5.38 | 33.33% | -1.87 |
| 4h8w | -1.33 | -12.71 | 50.00% | -1.84 | 10178.09 | -1.14 | 66.67% | -2.00 | 4219.10 | -0.12 | 16.67% | -2.00 | 1347.02 | 266.59 | 41.67% | -2.18 | 407.69 | 5.12 | 50.00% | -2.04 | 53.51 | -5.07 | 75.00% | -2.36 | 312.93 | -3.84 | 25.00% | -1.66 |
| 4ki5 | -8.15 | -16.58 | 26.67% | -1.80 | 2626.19 | 27.58 | 46.67% | -1.93 | 4943.25 | -0.55 | 46.67% | -1.93 | 3439.33 | -1.96 | 40.00% | -1.90 | 1027.67 | 3.06 | 40.00% | -1.79 | 74.95 | -5.23 | 73.33% | -2.17 | 318.38 | -0.54 | 33.33% | -1.72 |
| 4lvn | 40.37 | -11.59 | 46.15% | -3.05 | 5298.71 | -1.29 | 53.85% | -1.80 | 2298.31 | -2.59 | 38.46% | -1.80 | 1920.97 | 25.46 | 38.46% | -1.63 | 1481.29 | -5.19 | 69.23% | -2.36 | 254.52 | -4.81 | 76.92% | -2.17 | 1362.36 | -4.38 | 25.00% | -1.40 |
| 4ot1 | -11.19 | -25.77 | 41.67% | -2.50 | 5396.84 | 372.36 | 41.67% | -2.13 | 14659.75 | 500.72 | 66.67% | -2.13 | 14443.68 | 4020.89 | 41.67% | -2.18 | 1069.16 | -1.03 | 41.67% | -2.03 | 242.35 | -4.61 | 50.00% | -1.71 | 492.04 | 161.16 | 25.00% | -1.66 |
| 4qci | 14.31 | -3.35 | 53.85% | -2.61 | 1380.37 | 130.53 | 38.46% | -2.28 | 2667.14 | -0.28 | 38.46% | -2.28 | 1599.80 | -0.38 | 23.08% | -1.69 | 47.93 | -2.00 | 38.46% | -2.04 | 20.01 | -2.29 | 53.85% | -2.23 | 39.84 | -3.76 | 30.77% | -1.71 |
| 4xnq | -18.37 | -20.88 | 75.00% | -1.89 | 10639.18 | 2.11 | 56.25% | -2.24 | 5229.34 | 61.78 | 25.00% | -2.31 | 5711.22 | 123.66 | 25.00% | -1.93 | 714.34 | -0.51 | 43.75% | -2.33 | 70.23 | -4.44 | 68.75% | -2.60 | 311.06 | 3.14 | 18.75% | -1.64 |
| 4ydk | -30.59 | -35.64 | 36.36% | -2.55 | 14084.25 | -0.35 | 31.82% | -2.40 | 14107.70 | 17.93 | 31.82% | -2.47 | 8232.12 | 1800.22 | 40.91% | -1.40 | 1586.95 | 171.24 | 50.00% | -2.26 | 121.44 | -6.48 | 68.18% | -2.23 | 938.56 | 28.09 | 22.73% | -2.00 |
| 5b8c | -15.23 | -15.23 | 38.46% | -1.80 | 3575.88 | 34.60 | 46.15% | -1.57 | 3630.36 | -4.90 | 38.46% | -1.57 | 3476.96 | 141.64 | 30.77% | -1.66 | 171.90 | 10.37 | 36.84% | -1.70 | 40.35 | -6.90 | 76.92% | -1.63 | 312.28 | -6.37 | 38.46% | -0.87 |
| 5bv7 | 44.87 | -18.25 | 47.37% | -1.73 | 8940.31 | 428.40 | 36.84% | -1.44 | 7124.46 | 8.81 | 57.89% | -1.44 | 3557.20 | 81.48 | 47.37% | -2.30 | 502.22 | -3.18 | 55.56% | -2.10 | 88.84 | -7.49 | 73.68% | -2.04 | 381.55 | 28.72 | 21.05% | -1.25 |
| 5d93 | -6.28 | -8.44 | 44.44% | -2.28 | 7994.29 | 198.09 | 44.44% | -2.00 | 1327.34 | -2.50 | 44.44% | -1.60 | 91.41 | -2.97 | 55.56% | -1.89 | 70.24 | 83.93 | 35.29% | -2.07 | 18.64 | -4.53 | 66.67% | -2.12 | 91.64 | -4.49 | 33.33% | -1.36 |
| 5en2 | -20.91 | -8.44 | 35.29% | -1.95 | 10072.38 | 1401.50 | 47.06% | -1.71 | 5079.34 | 25.09 | 35.29% | -1.71 | 3919.12 | 26.12 | 41.18% | -1.76 | 521.48 | 147.83 | 53.33% | -2.19 | 118.27 | -6.66 | 64.71% | -2.11 | 404.68 | 53.20 | 23.53% | -1.66 |
| 5f9o | 5.96 | -16.16 | 46.67% | -2.71 | 5406.24 | 413.41 | 33.33% | -2.03 | 3770.48 | 622.27 | 40.00% | -1.45 | 4852.12 | -0.12 | 53.33% | -2.21 | 345.58 | -3.74 | 61.54% | -2.27 | 85.64 | -3.61 | 92.31% | -2.40 | 184.90 | -3.97 | 26.67% | -1.99 |
| 5ggs | -15.18 | -17.75 | 38.46% | -2.51 | 5621.88 | 36.53 | 46.15% | -2.14 | 4429.14 | 124.02 | 23.08% | -2.14 | 3285.12 | 930.05 | 30.77% | -1.68 | 206.95 | 0.53 | 54.55% | -2.22 | 34.25 | -8.49 | 92.31% | -2.18 | 138.10 | -9.29 | 46.15% | -1.55 |
| 5hi4 | 0.48 | -18.07 | 45.45% | -2.34 | 9789.87 | 13849.60 | 54.55% | -2.21 | 3004.20 | 402.13 | 18.18% | -2.21 | 991.65 | 588.91 | 54.55% | -1.64 | 186.35 | 0.30 | 46.67% | -1.86 | 89.25 | -5.44 | 72.73% | -1.83 | 377.22 | -3.98 | 27.27% | -1.13 |
| 5j13 | 0.61 | -18.76 | 63.64% | -2.06 | 8911.86 | 2045.54 | 73.33% | -1.86 | 2597.89 | -0.96 | 53.33% | -1.86 | 4840.53 | 4973.45 | 46.67% | -2.41 | 135.69 | -2.59 | 66.67% | -2.27 | 55.16 | -6.97 | 66.67% | -2.39 | 124.44 | -1.07 | 46.67% | -1.79 |
| 5l6y | -0.45 | -10.43 | 46.67% | -1.94 | 15653.52 | 127.28 | 53.33% | -2.21 | 2982.89 | -0.05 | 33.33% | -2.21 | 2987.61 | 1163.43 | 33.33% | -2.34 | 134.68 | -2.91 | 66.67% | -2.14 | 55.10 | -3.54 | 66.67% | -1.84 | 154.32 | -3.89 | 33.33% | -1.62 |
| 5mes | -16.95 | -7.42 | 66.67% | -2.39 | 7281.40 | 45.85 | 41.67% | -2.23 | 2132.88 | -2.61 | 41.67% | -2.23 | 1377.29 | 1421.52 | 58.33% | -1.91 | 186.88 | -2.91 | 58.33% | -2.05 | 28.39 | -4.54 | 75.00% | -2.31 | 211.29 | -3.27 | 33.33% | -1.35 |
| 5nuz | -16.73 | -18.19 | 23.08% | -2.16 | 11978.97 | -1.37 | 53.85% | -1.95 | 3901.04 | 80.63 | 38.46% | -1.95 | 1394.59 | 230.11 | 46.15% | -1.74 | 213.66 | 162.44 | 38.46% | -2.39 | 126.07 | -4.80 | 61.54% | -2.15 | 300.00 | -3.14 | 30.77% | -1.51 |

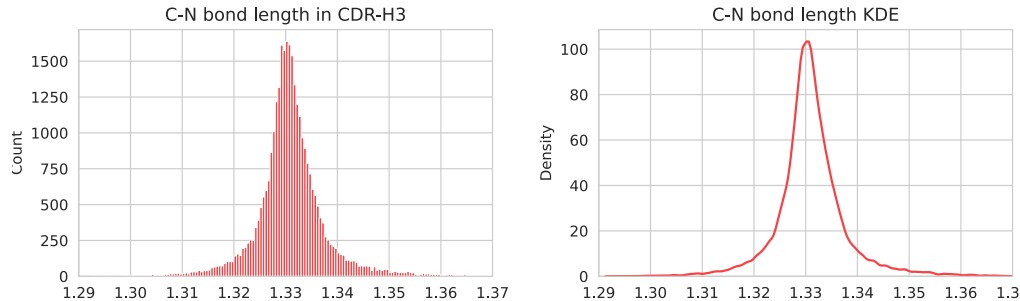

Figure 7: **Left**: the distribution of peptide bond length within CDR-H3 in the SAbDab dataset; **Right**: the kernel density estimation (KDE) function fit on the natural peptide bond length distribution.

## F   Arbitrary Preferences

### F.1   Incorporating Auxiliary Loss

A predominant advantage of the ABDPO is its unique capacity to seamlessly integrate traditional bioinformatics, computational biology, and computational chemistry tools — those incapable of directly computing gradients — into the training regimen of AI models. This integration significantly broadens the ABDPO's applicability and versatility in antibody design. However, it is pertinent to acknowledge the existence of antibody energies/properties for which gradient calculations are feasible. Indeed, fundamental geometric characteristics, such as bond lengths, angles, and torsion angles, alongside more intricate properties predicted by deep-learning models, are gradient-computable. These gradient-computable features offer an explicit direction for optimization, potentially enhancing the effectiveness and efficiency of the model optimization process.

In light of this, we initiated another experiment aimed at exploring ABDPO's compatibility with traditional gradient-based losses, extending beyond the DPO loss. Specifically, we propose a special version based on ABDPO+, ABDPO++, which incorporates an auxiliary loss about peptide bond length. As a covalent bond, the variation range of peptide bond lengths is very limited, and thus we can consider the length of peptide bonds to be a fixed value and then utilize an MSE loss to directly penalize the unreasonable peptide bond length in generated antibodies.

In practice, we consider the ground truth peptide bond length to be 1.3310 (the average length of peptide bonds within CDR-H3 in SAbDab, the distribution could be seen in Fig. 7 left) and apply the auxiliary loss only when the sampled t is near 0 ($t < 15$ in this experiment while $T$ is 100), and the weight is set to 0.25. The peptide bond length is calculated based on the predicted $(s_j^0, \mathbf{x}_j^0, \mathbf{O}_j^0)$ which is denoised with one step from $(s_j^t, \mathbf{x}_j^t, \mathbf{O}_j^t)$, then an MSE loss of peptide bond length can be calculated. Finally, this auxiliary loss, together with various DPO losses, updates the model through the conflict mitigation mentioned in Sec. 3.3.

Table 6: Summary of CDR $E_{\text{total}}$, CDR-Ag $\Delta G$ (kcal/mol), pLL, PHR, C-N$_{\text{score}}$, AAR, and RMSD of reference antibodies and antibodies designed by ABDPOw/O and baselines in the experiment involves auxiliary loss. ($\downarrow$) / ($\uparrow$) denotes a smaller / larger number is better.

| Methods | CDR $E_{\text{total}}$ ($\downarrow$) | CDR-Ag $\triangle G \downarrow$ | pLL ($\uparrow$) | PHR ($\downarrow$) | C-N$_{\text{score}}$ ($\uparrow$) | AAR ($\uparrow$) | RMSD ($\downarrow$) |
|---|---|---|---|---|---|---|---|
| HERN | 10887.77 | 2095.88 | -2.02 | 40.46% | 0.12 | 32.38% | 9.18 |
| MEAN | 7162.65 | 1041.43 | **-1.79** | **36.20%** | 1.68 | 36.30% | **1.69** |
| dyMEAN | 3782.67 | 1730.06 | -1.82 | 43.72% | 2.08 | **40.04%** | 1.82 |
| DiffAb | 1729.51 | 1297.25 | -2.10 | 41.27% | 3.85 | 34.92% | 1.92 |
| ABDPO | **629.44** | **307.56** | -2.18 | 69.67% | 2.55 | 31.25% | 1.98 |
| ABDPO+ | 1106.48 | 637.62 | -2.00 | 44.21% | 2.95 | 36.27% | 2.01 |
| ABDPO++ | 1349.39 | 747.89 | -1.99 | 44.46% | **4.51** | 36.30% | 1.95 |

To evaluate the consistency of generated antibodies' peptide bond length to the natural antibodies, we fit a Kernel Density Estimation function using the length of peptide bonds found within the CDR-H3 region of natural antibodies (shown in Fig. 7 right), then the density of the generated peptide bond length, C-N$_{score}$, is used to represent the consistency. We report the average experiment result in Tab. 6. It can be observed that ABDPO++ significantly optimized the length of the peptide bond, achieving the best C-N$_{score}$ of 4.51, while maintaining the optimization to the other 4 preferences. The experimental result demonstrates the compatibility of AbDPO with traditional gradient-based losses, indicating that AbDPO has a wider scope in actual application.

## F.2 Incorporating Energy Minimization

Energy minimization is indispensable in the standard protein design protocol and is typically applied to the raw co-crystal structure and the generated structure. Most existing AI-based antibody design methods have not undergone similar operations, but to verify the performance of ABDPO in a more realistic workflow environment, we have also proposed another version based on ABDPO+ that integrates energy minimization, ABDPOW/O.

For the minimization of the raw co-crystal structure, we compared the performance of baseline methods trained with and without minimized co-crystal structure but observed no significant difference. A possible reason for this is that most of the methods do not generate the side chain and thus are not sensitive to energy minimization, which mainly optimizes the side-chain conformation. Thus we follow the previous studies, and directly use raw co-crystal structure to train the baseline models and the pre-trained model in ABDPO.

We carry out minimization during the evaluation phase and apply the minimization to the generated antibodies before energy calculation. Therefore, the preference dataset used in ABDPOW/O is built upon the minimized energy. The energy minimization process consists of two parts, **peptide bond length rectification** and **loop refinement**. We first set the length of the peptide bond to 1.3310, the average length of the peptide bonds within CDR-H3 in the SAbDab dataset. Then we use *LoopMover_Refine_CCD* from pyRosetta to refine the structure of the designed CDR loop. To reduce time consumption in loop refinement, we set the *outer_cycles* to 1 and *max_inner_cycles* to 10 (a bigger number of cycles will lead to better energy performance undoubtedly, but also makes the time consumption uncontrollable).

Another modification of ABDPOW/O compared to ABDPO+ is that the decomposition of Res$_{CDR}$-Ag $\Delta$G into Res$_{CDR}$-Ag $E_{nonRep}$ and Res$_{CDR}$-Ag $E_{Rep}$ is canceled. Energy decomposition is indispensable in the main experiment because of the huge repulsion, and is not necessary in this experiment as the repulsion would be diminished by the post-minimization process.

Table 7: Summary of CDR $E_{total}$, CDR-Ag $\Delta G$ (kcal/mol), PHR, and pLL of reference antibodies and antibodies designed by ABDPOW/O and baselines in the experiment involves energy minimization. ($\downarrow$) / ($\uparrow$) denotes a smaller / larger number is better.

| Methods | CDR $E_{total}$ ($\downarrow$) | CDR-Ag $\Delta G$ ($\downarrow$) | PHR ($\downarrow$) | pLL ($\uparrow$) |
|---------|------|------|------|------|
| RAbD | -0.6699 | -10.2772 | 0.4578 | -2.2046 |
| HERN | 2765.5834 | 0.8332 | 41.41% | -2.0409 |
| MEAN | 1162.0961 | 0.0508 | **30.63%** | **-1.7936** |
| dyMEAN | 611.1203 | -2.051 | 43.73% | -1.8187 |
| DiffAb | 82.6216 | -0.2734 | 38.58% | -2.0963 |
| ABDPOW/O | **69.8181** | **-3.0007** | 36.71% | -2.0251 |

In Tab. 7, we report the average values of the evaluation metrics for all the generated antibodies in this experiment. Given that the peptide bond length has been rectified, measuring the C-N score is deemed unnecessary in this context. It can be observed that the post-minimization eliminates most of the clashes between the designed antibodies and the corresponding antigens, making CDR-Ag $\Delta G$ fall within a reasonable range of value. ABDPOW/O still achieves the best performance in the two energy-based metrics, CDR $E_{total}$ and CDR-Ag $\Delta G$, and surpasses DiffAb in all metrics. This experiment proves **(1)** the effectiveness of ABDPO in a more realistic setting, and **(2)** the ability of ABDPO to optimize the energies/properties not directly calculated from the generated antibodies. The values of the two sequence-related metrics, PHR and pLL, for the baseline methods slightly

differ from those in Tab. 1. This discrepancy arises because we imposed a maximum processing time during the loop refinement phase, leading to the exclusion of samples whose refinement was incomplete within the allocated time.

## G   Extended Ablation Studies

Due to the massive training cost in the RAbD benchmark, we investigate the effectiveness and necessity of each proposed component on five representative antigens, whose PDB IDs are 1a14, 2dd8, 3cx5, 4ki5, and 5mes. From the results in Fig. 8, it is clear that ABDPO can significantly boost the overall performance of ablation cases. Note that moving averages are applied to smooth out the curves to help in identifying trends, including Fig. 4. We present observations and constructive insights of the three proposed components as follows:

1. The residue-level DPO is vital for training stability specifically for CDR $E_{\text{total}}$. As aforementioned in Section 3.2, the residue-level DPO implicitly provides fine-grained and rational gradients. In contrast, vanilla DPO (without residue-level DPO) may impose unexpected gradients on stable residues, which incurs the adverse direction of optimization. According to each energy curve in Figure 8, we observe that residue-level DPO surpasses vanilla DPO by at least one energy term.

2. Without Energy Decomposition, all five cases appear undesired "shortcuts" aforementioned in Section 3.3. We observe that the energy of CDR $E_{\text{total}}$ exhibits a slight performance improvement over the ABDPO after the values of attraction and repulsion reach zero. We suppose that is the result of the combined effects of low attraction and repulsion. Because the generated CDR-H3 is far away from the antigen in this case, the model can concentrate on refining CDR $E_{\text{total}}$ without the interference of attraction and repulsion.

3. The Gradient Surgery can keep a balance between attraction and repulsion. We can see the curves of $E_{\text{nonRep}}$ are consistently showing a decline, while the curves of $E_{\text{Rep}}$ are showing an increase. This observation verifies that ABDPO without Gradient Surgery is unable to optimize $E_{\text{nonRep}}$ and $E_{\text{Rep}}$ simultaneously. Additionally, the increase in attraction significantly impacts the repulsion, causing the repulsion to fluctuate markedly.

## H   Limitations and Future Work

**Diffusion Process of Orientations**   As Luo et al. [36] stated and we have mentioned in Sec. 3.1, Eq. (1) is not a rigorous diffusion process. Thus the loss in Eq. (7) cannot be rigorously derived from the KL-divergence in Eq. (4), though they share the idea of reconstructing the ground truth data by prediction. However, due to the easy implementation and fair comparison with the generative baseline, i.e., DiffAb [36], we adopt Eq. (7) in the ABDPO loss in Eq. (8). In practice, we empirically find that it works well. FrameDiff [50], a protein backbone generation model, adopts a noising process and a rotation loss that are well compatible with the theory of score-based generative models (also known as diffusion models). In the future, we modify the diffusion process of orientations as Yim et al. [50] for potential further improvement.

**Energy Estimation**   In this work, we utilize Rosetta/pyRosetta to calculate energy, although it is already one of the most authoritative energy simulation software programs and widely used in protein design and structure prediction , the final energy value is still difficult to perfectly match the actual experimental results. In fact, any computational energy simulation software, whether it is based on force field methods such as OpenMM [14] or statistical methods like the Miyazawa-Jernigan potential [38], will exhibit certain biases and cannot fully simulate reality. Sometimes there is a significant difference between the energy calculated by the software and the results observed experimentally. One possible reason is that theoretical calculations often rely on the designed sequence and structure of antibodies; meanwhile, in actual experiments, the actual folding of the CDR region into the designed structure can be difficult, which leads to significant discrepancies in theoretical calculations. An in vitro experiment is the only way to verify the effectiveness of the designed antibodies. However, due to the significant amount of time consumed by in vitro experiments and considering that the main goal of our work is to propose a novel view of antibody design, we did not perform the in vitro experiment.

**Future Work on Preference Definition**   The preferences used in ABDPO determine the tendency of antibody generation, and we will strive to continue exploring the definition of preference to more

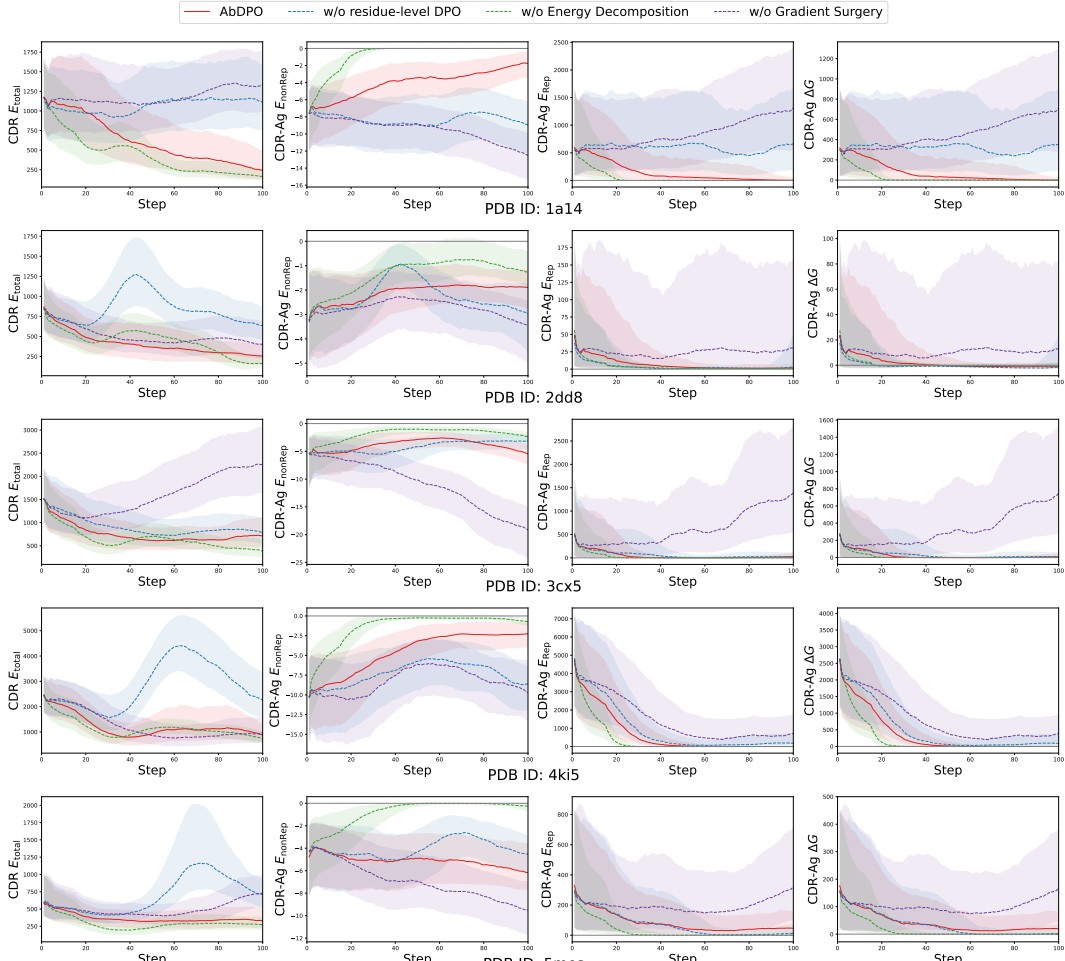

Figure 8: Changes of median CDR $E_{\text{total}}$, CDR-Ag $E_{\text{nonRep}}$, CDR-Ag $E_{\text{Rep}}$, and CDR-Ag $\Delta G$ (kcal/mol) over-optimization steps, shaded to indicate interquartile range (from 25-th percentile to 75-th percentile). The rows represent PDB 1a14, 2dd8, 3cx5, 4ki5, and 5mes respectively, in a top-down order.

closely align the antibody design process with the real-world environment of antibody activity. Further, we aim to synchronize the preference with the outcomes of in vitro experiments and expect that our method will ultimately generate effective antibodies in real-world applications. The exploration of preference can be divided into two aspects: enhancing existing preferences and integrating new components or energies.

1. The improvement to current preference: (1) performing more fine-grained calculations on the current three types of energy, such as decomposing CDR $E_{\text{total}}$ into interactions between the CDR and the rest of the antibody, interactions within the CDR, and energy at the single amino acid level; (2) exploring the varying importance of preferences for antibodies and determining the relative weights of each preference during the optimization and ranking of generated antibodies.

2. The incorporation of new components or energies is intended to address additional challenges in antibody engineering, focusing on aspects such as antibody stability, solubility, immunogenicity, and expression level. Additionally, we consider integrating components that target antibody specificity.

# I  Potential Societal Impacts

Our work on antibody design can be used in developing potent therapeutic antibodies and accelerate the research process of drug discovery. The generality of our method extends beyond its current application; it is adaptable for various computer-aided design scenarios including, but not limited to, small molecule, material, and chip design. It is also needed to ensure the responsible use of our method and refrain from using it for harmful purposes.

