# OpenReview forum: "Antigen-Specific Antibody Design via Direct Energy-based Preference Optimization"
_NeurIPS.cc/2024/Conference — NeurIPS 2024 poster_

### Official Review · Reviewer_d9wd · 2024-07-07

**Soundness:** 3
**Presentation:** 3
**Contribution:** 3
**Rating:** 7
**Confidence:** 4

**Summary:**

This paper uses a pre-trained conditional diffusion model for antibody design. This diffusion model is fine-tuned using a direct energy-based preference optimization method, focusing on optimizing residue-level energy preferences to enhance the generation of antibodies with desirable structures and high binding affinities. The authors also compared their method with SOTA baselines on 55 cases from RAbD benchmarks and showed good results.

**Strengths:**

1. The proposed algorithm is novel and interesting. The authors combine diffusion models and DPO to solve the specific problem of antibody design.
2. The algorithm is clear. The authors both define the diffusion process and DPO formation with very clear definitions. Meanwhile, all the figures with protein structure are very clear and informative.
3. The results are convincing. AbDPO archives good results on CDR total energy compared to other methods.
4. The experiments part is very detailed. The authors conduct experiments on 55 of 60 antibodies on both their methods and the compared baselines.

**Weaknesses:**

Lack of explanation of SE(3)-equivariant neural network. The author uses the diffusion model with such an equivariant neural network from Luo et al. Lacking such an explanation may hurt the understanding of the whole method.

**Minor:**
- There are some writings inconsistent. For example, equations under line 130 are not marked with numbers, but equations under line 138 have numbers. The ABDPO in everywhere this paper is written as \textsc{AbDPO}, but in line 242 is "ABDPO". In line 235, the first letter in pyRosetta should be capitalized since it is a proper noun. Checking these format issues will improve the consistency in the future published version.

Reference:
----------------------------------
Luo et al: Shitong Luo, Yufeng Su, Xingang Peng, Sheng Wang, Jian Peng, and Jianzhu Ma. 2022. Antigen-Specific Antibody Design and Optimization with Diffusion-Based Generative Models

**Questions:**

1. Why we need to use SE(3)-equivariant neural network?
2. What is the inference time for generating one sample or one batch?

**Limitations:**

Limitations are adequately addressed.

---

> ### Author Rebuttal · Authors · 2024-08-07
>
> Thank you for your strong support! Please see below for our responses to the comments.
>
> **Q1: Lack of explanation of SE(3)-equivariant neural network. The author uses the diffusion model with such an equivariant neural network from Luo et al. Lacking such an explanation may hurt the understanding of the whole method.**
>
> A1: Thanks for your valuable suggestion. We will complete the explanation in the revision.
>
> **Q2: There are some writings inconsistent.**
>
> A2: Thanks for your careful inspection, we will fix it in the revision.
>
> **Q3: Why we need to use SE(3)-equivariant neural network?**
>
> A3: SE(3)-equivariance is a crucial property in protein design because the structure of proteins is independent to the observation view. Utilizing SE(3)-equivariance can ensure stable and predictable performance in the presence of nuisance transformations of the data input[1]. In addition, we chose to build our method on DiffAb, which uses equivariant neural networks. Other baselines, such as MEAN and dyMEAN, are also E(3)-equivariant. Therefore, we retained the equivariant neural networks to ensure a fair comparison. Nevertheless, we would like to emphasize that our proposed method is not specific to equivariant NNs and can be applied to other base models beyond equivariant NNs.
>
> Reference:
>
> [1] Fuchs, Fabian, Daniel Worrall, Volker Fischer, and Max Welling. "Se (3)-transformers: 3d roto-translation equivariant attention networks." Advances in neural information processing systems 33 (2020): 1970-1981.
>
> **Q4: What is the inference time for generating one sample or one batch?**
>
> A4: Our model could be regarded as an aligned model to the specific preference of a pre-trained model, so the inference time will not be changed. We record the detailed time cost in each antigen-antibody complex and show them below (batch_size=16, single A100 40G GPU).
> The inference time consists of two parts: model inference time and .pdb file reconstruction time. The inference time is related to data length. Most antigen-antibody complexes are truncated to 256 residues when fed into the model, and the corresponding inference time is around 16-17s. The reconstruction time is determined by the total number of residues in the complex. For the complex with 500 residues, the  reconstruction time of one batch of samples is around 2s. We list the detailed time cost below (the last few rows are dropped due to the limitation of space)
>
> | pdb_id | infer_time (s) | data_length | reconst_time (s) | pdb_length |
> |--------|----------------|-------------|------------------|------------|
> | 1a14   | 16.73          | 256         | 2.36             | 612        |
> | 1a2y   | 16.76          | 256         | 1.29             | 351        |
> | 1fe8   | 16.86          | 256         | 2.28             | 610        |
> | 1ic7   | 16.76          | 256         | 1.25             | 349        |
> | 1iqd   | 16.73          | 256         | 2.09             | 563        |
> | 1n8z   | 16.75          | 256         | 3.6              | 1015       |
> | 1ncb   | 16.75          | 256         | 3.28             | 823        |
> | 1osp   | 16.73          | 256         | 2.44             | 682        |
> | 1uj3   | 16.88          | 256         | 2.44             | 635        |
> | 1w72   | 16.73          | 256         | 2.81             | 707        |
> | 2adf   | 16.73          | 256         | 2.28             | 615        |
> | 2b2x   | 16.71          | 256         | 2.19             | 607        |
> | 2cmr   | 16.8           | 256         | 2.35             | 604        |
> | 2dd8   | 16.81          | 256         | 2.49             | 623        |
> | 2vxt   | 16.75          | 256         | 2.21             | 575        |
> | 2xqy   | 16.72          | 256         | 3.26             | 897        |
> | 2xwt   | 16.76          | 256         | 2.43             | 661        |
> | 2ypv   | 16.75          | 256         | 2.67             | 655        |
> | 3bn9   | 16.81          | 256         | 2.45             | 662        |
> | 3cx5   | 16.8           | 256         | 1.74             | 418        |
> | 3ffd   | 6.44           | 144         | 1.79             | 444        |
> | 3hi6   | 16.78          | 256         | 2.34             | 606        |
> | 3k2u   | 16.81          | 256         | 2.46             | 655        |
> | 3l95   | 16.77          | 256         | 3.5              | 661        |
> | 3mxw   | 16.76          | 256         | 2.26             | 585        |
> | 3nid   | 16.74          | 256         | 3.33             | 883        |
> | 3o2d   | 16.72          | 256         | 2.26             | 615        |
> | 3rkd   | 16.73          | 256         | 2.21             | 577        |
> | 3s35   | 14.93          | 240         | 2.14             | 539        |
> | 3w9e   | 16.74          | 256         | 2.4              | 654        |
> | 4cmh   | 16.73          | 256         | 2.41             | 668        |
> | 4dtg   | 10.09          | 192         | 1.87             | 499        |
> | 4dvr   | 16.71          | 256         | 2.74             | 718        |
> | 4ffv   | 16.79          | 256         | 4.74             | 1144       |
> | 4fqj   | 16.72          | 256         | 2.68             | 731        |
> | 4g6j   | 16.81          | 256         | 2.19             | 578        |
> | 4g6m   | 16.85          | 256         | 2.31             | 583        |
> | 4h8w   | 16.76          | 256         | 2.82             | 760        |
> | 4ki5   | 16.77          | 256         | 2.23             | 587        |
> | 4lvn   | 16.79          | 256         | 2.99             | 758        |
> | 4ot1   | 15.79          | 248         | 2.08             | 558        |
> | 4qci   | 13.17          | 224         | 1.94             | 511        |
> | 4xnq   | 16.84          | 256         | 2.39             | 627        |
> | 4ydk   | 16.86          | 256         | 2.94             | 785        |
> | 5b8c   | 15.83          | 248         | 1.32             | 346        |
> | 5bv7   | 16.8           | 256         | 3.2              | 809        |

---

> > ### Comment · Reviewer_d9wd · 2024-08-07
> >
> > Thanks for your response! I hope to see the revised version of this paper later.
> > Please remember to add an explanation of the SE(3) network in your paper, either in the main paper or appendix. That would be more helpful in understanding the paper.

---

### Official Review · Reviewer_FkeK · 2024-07-10

**Soundness:** 3
**Presentation:** 2
**Contribution:** 2
**Rating:** 5
**Confidence:** 3

**Summary:**

This paper proposes a new perspective for antibody design-- incorporating the energy factors aiming to minimize the overall energy of designed sequence and structure. It involves diffusion to maintain the sequence-structure co-design. Towards the variance of energy factors, the paper proposes the idea of gradient surgery to simplify it. The paper conducted extensive experiments, showing its performance towards other baselines. The paper is easy to follow.

**Strengths:**

* The paper proposes an energy-based preference optimization approach to achieve better rationality and binding affinity, which is inspiring in the field of antibody design.
* The paper decomposites the energy factors, simplifying the gradient calculation process.
* The paper achieved significantly lower energy than other approaches, reflecting the effectiveness of the model design.
* The paper clearly states why it chooses energy as the main evaluation metric with strong statistical evidence.
* The paper provides a detailed sample-level comparison of existing baseline approaches, which is a solid solution and beneficial to successive works.

**Weaknesses:**

* Although the authors stated why they chose energy as the main metric, the AAR is about 10% lower than dyMEAN. The authors did not provide a solid reason for what caused the result (Are all the lower results considered hacked or biased? Maybe a sample-level analysis would be more convincing.)
* The derivation of the final loss for fine-tuning could be simplified.
* The adaptation of RLHF into this approach should be further clarified.

**Questions:**

* This paper relies heavily on energy function and sidechain packing methods in pyrosetta. Would it be helpful to directly train the diffusion model from scratch by optimizing the loss function using pyrosetta?
* There are several other tasks except for CDR-H3 design, as the paper aims to optimize the energy of the complex, would it be helpful to test the method with other downstream tasks that may be closely related to the energy(refer to dyMEAN)?

**Limitations:**

As stated in the article.

---

> ### Author Rebuttal · Authors · 2024-08-07
>
> Thank you for your helpful feedback! We address your questions as follows.
>
> **Q1: AAR is about 10% lower than dyMEAN. The authors did not provide a solid reason for what caused the result.**
>
> A1: As mentioned in Appendix A, AAR is easily hacked and conceals numerous issues. Certain intrinsic patterns within CDR-H3 sequences enable the model to achieve a seemingly satisfactory AAR by memorizing these patterns (also noted in the Limitations section of dyMEAN). However, this leads the model to produce highly similar sequences for all antigens, which is highly impractical. The possible reason is the severe lack of data, causing the model to merely learn the marginal distribution of residue types at each position. dyMEAN is a typical example of this scenario. To illustrate the sequences generated by dyMEAN with over 40% AAR, we reproduced dyMEAN, generated CDR-H3 for 55 antigens, and visualized the sequences. As shown in Fig. 2 in the uploaded pdf, through alignment (aligned and visualized by MAFFT[1]), it is evident that dyMEAN generates almost identical sequences, while AbDPO+ is considerably better. In this case, AAR is not a meaningful measure.
>
> As for why AbDPO did not achieve a high AAR, we believe it is due to the different learning objectives between AbDPO and the baselines. The learning objective of baselines is to generate antibodies consistent with natural antibodies in terms of sequence and structure. Therefore, they perform better on metrics like AAR, which measures similarity to natural ones. However, AbDPO's learning objective is to better meet specific preferences, while not optimizing for consistency with natural ones. Consequently, its AAR is lower than dyMEAN. Nevertheless, sequence pLL leads to a significant improvement in AAR (31.25% to 36.27%) when used as an optimization objective.
>
> Reference:
>
> [1] Madeira, Fábio,et al. "The EMBL-EBI Job Dispatcher sequence analysis tools framework in 2024." Nucleic Acids Research.
>
> **Q2: The adaptation of RLHF into this approach should be further clarified.**
>
> A2: We can consider the antibody generation model obtained through conventional training methods as a pre-trained model. However, this pre-trained model does not meet our requirements, such as physical energy. This situation is similar to the challenges faced in current generative AI, where pre-trained models often require further alignment to match human preferences. Therefore, we can leverage methods like RLHF to align the antibody generation models with the desired properties, essentially optimizing the model. In AbDPO, we optimize the pre-trained model using such methods, allowing us to generate antibodies that satisfy multiple preferences.
>
> **Q3: Would it be helpful to directly train the diffusion model from scratch by optimizing the loss function using pyrosetta?**
>
> A3: Directly training the diffusion model from scratch is not practical.
>
> The reasons are twofold:
>
> 1. Since the loss function using pyrosetta is not differentiable, RL algorithms can be applied, such as policy gradient methods. In this case, the antibody design problem is formulated as a high-dimensional non-linear decision making problem. The search space for this problem is large and involves both continuous and discrete variables. Directly training from scratch is hard due to its instability. Specifically, useless efforts might be made, especially at the beginning of the training, when the policy can only generate samples with unsatisfactory rewards. A feasible solution to address this challenge is to use a pre-trained policy that can significantly prune the search space and accelerate the convergence rate, which has been validated on many applications, such as [1] and [2]. And this solution is the pretraining-finetuning paradigm that we utilized in our work. The pre-trained diffusion model can be viewed as an expressive policy.
>
> 2. Another reason is that low pyrosetta energy is not our only purpose. We aim to generate valid antibodies with low energy. Some intricate properties cannot be explicitly formulated as rewards but can be learned from data by generative modeling. Our optimization objective is equivalent to maximizing rewards with regularization with reference model. The regularization term keeps knowledge learned from data, which cannot be achieved by training from scratch by optimizing the energy function.
>
> References:
>
> [1] Silver, David, et al. Mastering the game of go without human knowledge. Nature
>
> [2] Haarnoja, Tuomas, et al. Soft actor-critic: Off-policy maximum entropy deep reinforcement learning with a stochastic actor. ICML.
>
> **Q4: Would it be helpful to test the method with other downstream tasks that may be closely related to the energy**
>
> A4: We test our method on the affinity optimization. Specifically, we test AbDPO, along with its counterpart, DiffAb by evaluating the best $\Delta\Delta G$ against the reference antibodies among 300 generated samples using [1] on 16 randomly selected antigens in the test set. **Note that we do not use the metric as the signal in fine-tuning**. The results are shown as follows:
>
> |pdb id|DiffAb|AbDPO|
> |--|--|--|
> |1a14|-3.91|**-5.46**|
> |1n8z|-5.23|**-6.80**|
> |1w72|0.34|**-1.82**|
> |2cmr|-3.12|**-3.62**|
> |3bn9|-2.44|**-4.42**|
> |3mxw|-4.12|**-5.45**|
> |3rkd|-6.21|**-6.99**|
> |4dvr|-5.28|**-5.77**|
> |4fqj|**-4.57**|-4.39|
> |4g6m|-5.94|**-5.97**|
> |4ki5|-3.17|**-4.83**|
> |4xnq|-6.78|**-7.71**|
> |5bv7|-6.19|**-7.66**|
> |5en2|-3.33|**-5.54**|
> |5f9o|-5.10|**-5.32**|
> |5nuz|-2.49|**-5.28**|
>
> Our results significantly outperform DiffAb, which shows the generalizability on other tasks. *We believe that our method can perform even better if we consider the metric itself, ie., predicted $\Delta\Delta G$ against the reference antibodies, in our preference definition when fine-tuning*.
>
> Reference:
>
> [1] Shan Sisi, et al. "Deep learning guided optimization of human antibody against SARS-CoV-2 variants with broad neutralization." Proceedings of the National Academy of Sciences

---

> > ### Comment · Reviewer_FkeK · 2024-08-08
> >
> > Thanks for the detailed rebuttal. My major questions have been solved and I increased the score.

---

### Official Review · Reviewer_X6SE · 2024-07-11

**Soundness:** 2
**Presentation:** 3
**Contribution:** 2
**Rating:** 6
**Confidence:** 4

**Summary:**

The paper proposes an approach for fine-tuning diffusion models for the design of antibodies. The core diffusion-based generative model comes from Luo et al. [36] and to my understanding there are no technical changes to it. The second component is direct preference based optimization, inspired by fine-tuning of large language models. This is the main technical contribution and builds entirely on the works by Rafailov et al [41] and Wallace et al [46]. The reward signal comes from binding free energy that is decomposed at the residue level. Different components include attraction and repulsion forces that can be linked to antibody function. To overcome possibly diverging gradients associated with different energy components, the authors propose to leverage gradient surgery from Yu et al [51] that essentially increases cosine similarity between gradients for different tasks/energy components.

Empirical evaluation is focused on qualitative aspects and whether the approach is able to discover better binders than initially given complexes, measured using binding free energy.

**Strengths:**

I think it is an interesting approach to merry molecular simulations and physics based energy calculations with diffusion and generative models. Especially given the small number of available crystal structures in the SAbDAB database.

Adding energy-based signal via direct preference optimization is an interesting re-purposing of that method.

Empirical evaluation goes beyond amino-acid recovery rate and RMSD metrics. The “success” at generating better binders quantified via improvement in binding free energy relative to the initial complex is an interesting metric.

**Weaknesses:**

Table 1 indicates that the approach is able to design better binders, measured via binding free energy relative to the initial complex. However, this comes at the expense of increasing the number of hydrophobic residues which is typically associated with non-specific binding. This would in all likelihood be useless binders and from the perspective of function no better than baselines.

It would be interesting to see the results of a baseline that “fine-tunes” relative to ddG score directly. In my understanding, the results in Table 1 are vanilla baselines. None of them (e.g., MEAN, dyMEAN, HERN) has for instance been used in combination with iterative improvement algorithm and some physics based simulator. How would this compare to fine-tuning relative to the directed preference optimization?

It would be interesting to see the results relative to different physics-based simulators. I’m not sure how many different simulators were used to generate “fine-tuning” signal?

Would it be possible to include additional metrics such as lDDT, TM, and some metrics characterizing the fit on angles?

**Questions:**

Could you also list how many mutations from the original CDR sequence are the improved binders (Table 1)?

Can you share the structural similarity scores between epitopes in the train and test folds?

If you limit the fraction of hydrophobic residues (PHR) to the value for the original complex and accept only designs with lower score, how does Table 1 look? Can you list relative to how many test fold complexes each method achieves better binding free energy (e.g., MEAN wins on N, AbDPO wins on 20, etc.)?

How does Table 1 look like if you apply ITA to the vanilla baselines?

---

> ### Author Rebuttal · Authors · 2024-08-07
>
> Thank you for your detailed feedback. Please see below for our responses to the comments.
>
> **Q1: The approach can design better binders, but comes at the expense of increasing the number of hydrophobic residues which is typically associated with non-specific binding.**
>
> A1: This is exactly why we included PHR as an optimization objective in AbDPO+. We will show the performance of each method when limiting the PHR in A6, and AbDPO still performs better.
>
> **Q2: It would be interesting to see the results relative to different physics-based simulators. I’m not sure how many different simulators were used to generate “fine-tuning” signal?**
>
> A2: All the three energy signals were calculated using Rosetta, which is capable of both energy calculation and side-chain packing, and is widely used by researchers. In early stages, we also tried using OpenMM for energy calculations and obtained consistent results.
> Additionally, we used OpenMM to calculate the potential energy of the antigen-antibody complex where CDR-H3 is generated by AbDPO fine-tuned according to Rosetta energy. For 6 complexes tested, AbDPO achieved lower energy in 5 compared to DiffAb. (the remaining 49 complexes raise errors in OpenMM workflow when adding Hydrogens, and are not influential in Rosetta workflow that does not require Hydrogens)
>
> **Q3: Would it be possible to include additional metrics such as lDDT, TM, and some metrics characterizing the fit on angles?**
>
> A3: For the antibodies generated by all methods, we measured both the TM-score[1] on the heavy chain (Hc_TM-score) and the torsion performance (Torsion-score) of residues on the CDR-H3. For Hc_TM-score, there is not much difference in methods other than HERN. The Torsion-score is derived from a two-dimensional KDE function, which is based on the joint distribution of Phi and Psi torsions observed in natural CDR-H3s. Diffusion-based methods, DiffAb and AbDPO, perform significantly better than other methods.
>
> | |Hc_TM-score|Torsion_score|
> |--|--|--|
> |HERN|0.92|0.14|
> |MEAN |0.98|0.34|
> |dyMEAN|0.98|0.68|
> |DiffAb|0.97|0.98|
> |AbDPO|0.97|0.86|
> |AbDPO+|0.98|0.90|
> |AbDPO++|0.97|0.89|
>
> Reference:
>
> [1] Zhang, Yang, and Jeffrey Skolnick. "TM-align: a protein structure alignment algorithm based on the TM-score." Nucleic acids research.
>
> **Q4: Could you also list how many mutations from the original CDR sequence are the improved binders (Table 1)?**
>
> A4: Yes, we calculate the minimum number of mutations of the generated samples that exhibit energy levels close to the natural samples (i.e., the "successful" samples in Table 1) and the corresponding CDR lengths.
>
> |pdb_id|CDR_length |N_mut_min|
> |--|--|--|
> |1a14|15|7|
> |1ic7|7|3|
> |1iqd|10|7|
> |2b2x|12|6|
> |2dd8|11|3|
> |3bn9|9|8|
> |4ffv|10|6|
> |4qci|13|8|
> |5d93|9|6|
>
> **Q5: Can you share the structural similarity scores between epitopes in the train and test folds?**
>
> A5: We follow the metric widely used in protein structure design tasks to calculate the structural similarity scores. The similarity between two protein sets is defined as the average TM-score of each sample in one set to its most similar protein in another set. The similarity between the train and test folds is 0.6965. Following MEAN, the epitope is defined as the 48 residues of the antigen closest to the paratope.
>
> Additionally, we believe there is a misunderstanding here. The training data only appears during the pre-training phase, while in the optimization phase, we only used synthetic data.
>
> **Q6: If you limit the fraction of hydrophobic residues to the value for the original complex and accept only designs with lower score, how does Table 1 look?**
>
> A6: We perform an evaluation only on the samples that contain hydrophobic residues not exceeding the natural one, and the results are shown below. It can be seen that under this setting, the energy performance of almost all methods has deteriorated, but AbDPO and AbDPO+ still perform the best in terms of the two energies.
>
> | | AAR| CDR $E_{\text {total}}$ | CDR-Ag$\Delta G$ | pLL| PHR|
> |--|--|--|--|--|--|
> |HERN| 32.30% |11953.56|1949.35|-1.95|25.81%|
> |MEAN| 37.42% |8127.87|1412.02|-1.93|23.85%|
> |dyMEAN| 38.33%|6253.15|2906.5|-1.94|31.84%|
> |DiffAb|34.47%|2129.9|1646.6|-2.14|27.22%|
> |AbDPO|31.24%|907.18|453.71|-2.25|31.24%|
> |AbDPO+|32.91%|1464.53|815.14| -2.09|32.91%|
>
> **Q7: Can you list relative to how many test fold complexes each method achieves better binding free energy?**
>
> A7: Sure. For each complex, if a method achieves the best performance, it was considered a "win" for that method on that specific complex. We record the number of "win" complexes, N_win. We also observe that there are instances where none of the samples have a PHR lower than that of the natural sample. Therefore, we also list the number of complexes, denoted as N_phr, for which the method can generate samples with a PHR not exceeding that of the natural sample. The specific results are as follows:
>
> | | N_win | N_phr |
> |--|--|--|
> |HERN|8|35|
> |MEAN|2|32|
> |dyMEAN|3 | 12|
> |DiffAb| 1 | 38|
> |AbDPO| 20 | 36|
> |AbDPO+| 18| 52|
>
> **Q8: How does Table 1 look like if you apply ITA to the vanilla baselines?**
>
> A8: We follow the ITA setting from the MEAN codebase and implement ITA for MEAN, dyMEAN, and DiffAb. We adapt $\text{CDR}E\_{\text{total}}$ and $\text{CDR-Ag}\Delta G$ as two targets to gather high-quality candidates for each antibody. We observe that directly generating high-quality candidates can negatively affect other preferences. This further demonstrates the effectiveness of AbDPO in optimizing multiple objectives simultaneously.
>
> |  | AAR | RMSD | CDR $E_{\text {total}}$ | CDR-Ag$\Delta G$ | pLL | PHR | N_success|
> |--|--|--|--|--|--|--|--|
> | MEAN   |32.56% |2.19| 4731.58 |  363.26 |-1.83 |75.12% |0 |
> | dyMEAN |39.69% |2.00|6105.83 |1665.72 |-1.63 |42.09% |0 |
> | DiffAb |35.93% | 2.15 |1288.67 |814.93 |-1.82 |59.57% |0 |
> | ABDPO| 31.25% |1.98 |629.44 |307.56 |-2.18 |69.67% |9 |
> | ABDPO+ | 36.27% |2.01 |1106.48 |637.62|-2.00 |44.21% |5 |

---

> > ### Comment · Reviewer_X6SE · 2024-08-07
> > **Response to the rebuttal**
> >
> > I’m satisfied with the response and will be increasing my score. Well done!

---

### Official Review · Reviewer_i9FY · 2024-07-14

**Soundness:** 3
**Presentation:** 3
**Contribution:** 3
**Rating:** 5
**Confidence:** 3

**Summary:**

This paper applies direct preference optimization to antibody design. Specifically, it uses Rosetta binding energy to guide a pre-trained diffusion model to generate antibody CDR structures with low binding energy.

**Strengths:**

* Optimizing antibody binding energy is an important problem.
* The proposed gradient surgery procedure is technically interesting.

**Weaknesses:**

* To evaluate binding energy using Rosetta, it is necessary to run side-chain packing and energy minimization to clean up the predicted structure. Therefore, it usually takes couple of minutes to evaluate binding energy using Rosetta for just one structure. In other words, it is computationally expensive to guide diffusion models using Rosetta.
* Rosetta side-chain packing is stochastic and non-deterministic. Therefore, if we relax the structure generated by diffusion model multiple times, the calculated Rosetta energy will be very different, and the standard deviation can be very high (sometimes it can be twice or three times higher than the mean). In other words, it is very tricky to construct a preference dataset because you need to compare the binding energy distribution between two CDR sequences, and their standard deviation is very high.
* Despite the high standard deviation, the reported binding energy in this paper does not have standard deviation. It seems like the authors only calculate Rosetta binding energy once for each structure instead of running side-chain packing or energy minimization multiple times and take the average. Therefore, the reported results may not be statistically significant.

**Questions:**

* Can you report your model performance by running Rosetta relaxation multiple times with different random seeds and report standard deviation of the reported binding energy for each method?

**Limitations:**

Yes

---

> ### Author Rebuttal · Authors · 2024-08-07
>
> Thank you for your feedback. Please see below for our responses to the comments.
>
> **Q1: It is computationally expensive to guide diffusion models using Rosetta.**
>
> A1: Yes, the computational expense is indeed a drawback of Rosetta. We are aware of Rosetta's limitations and have discussed them in the appendix. However, despite these issues, the time cost of optimizing AbDPO for a single antigen is less than a day. Additionally, AbDPO is not specific to Rosetta. For example, other packing tools (like DiffPack) and energy calculation methods (such as OpenMM, which is significantly faster than Rosetta) can also be used. In fact, AbDPO supports optimization of any type of property, and we chose Rosetta because it can perform both packing and energy calculations and is widely used.
>
> **Q2: Rosetta side-chain packing is stochastic and non-deterministic. Therefore, if we relax the structure generated by diffusion model multiple times, the calculated Rosetta energy will be very different, and the standard deviation can be very high (sometimes it can be twice or three times higher than the mean). In other words, it is very tricky to construct a preference dataset because you need to compare the binding energy distribution between two CDR sequences, and their standard deviation is very high.**
>
> A2: We recognize that the same CDR-H3 (with the same sequence and structure) may result in different side-chain conformations after packing, and there will be energy differences between these side-chains. We observed this in our early exploration but still only performed the side-chain packing for each sample once. The reasons are as follows :
> - The differences between different samples generated by existing methods (such as DiffAb) are generally much larger than the energy differences between different side chain conformations of the same sample. For natural antibodies, total energy is relatively low due to their reasonable structure, so different side chains may have a significant impact on total energy. However, for generated antibodies with several clashes, the influence of side-chain conformations is not significant as the total energy could be extremely high.
> - Repeating the packing process on the same sample can indeed yield more accurate energy labels, but it will consume more time and we do not want the excellent performance of AbDPO to be based on a large amount of meticulously selected data. Although sampling only once may lead to incorrect judgments of win-lose relationships between two samples, this usually occurs when the two samples are quite similar. As long as the better sample has a higher probability of being sampled at lower energy, the model will generally be optimized toward generating antibodies with lower energy.
> - DPO is derived based on the Bradley-Terry model. In the Bradley-Terry model, the labels of winner and loser are not deterministic but exist in probabilistic form. We perform packing for each sample only once, which is similar to sampling winner and loser once. This is also theoretically justifiable.
>
> Furthermore, we use 1a14 as an example and sample 100 synthetic data from the 1a14 training dataset. For each synthetic sample, we repeat packing 128 times with different random seeds. Then we can compare the deviation between different synthetic data and different sidechains from the same synthetic data. The results are shown in Fig. 1 in the uploaded pdf, and the deviation between different synthetic data (the red line) is far greater than the deviation between different sidechains from the same synthetic data (the blue violin plot indicates the distribution of deviation within each synthetic data). This further supports our justification.
>
> **Q3: Can you report your model performance by running Rosetta relaxation multiple times with different random seeds?**
>
> A3: We repeat the optimization process 32 times with different random seeds for each AbDPOw/O generated sample. The average $\text{CDR} E_{\text{total}}$ and $\text{CDR-Ag} \Delta G$ of all generated antibodies are 65.19 and -2.67, which are slightly different from the reported values of 69.82 and -3.00.  Therefore, the experiment results reported in our paper are reliable. The reason we did not report the averaged results over multiple runs is due to time cost considerations.
>
> **Q4: Can you report standard deviation of the reported binding energy for each method?**
>
> A4: Thanks for your reminder, we provide the standard deviation of two energies for each method below. It could be seen that significant deviations are commonly present in the results generated by each method, and the deviation of AbDPO is smaller than that of DiffAb.
>
> |  | CDE$E_{\text{total}}$ avg | CDE$E_{\text{total}}$ std | CDR-Ag$\Delta G$ avg | CDR-Ag$\Delta G$ std |
> |----------|-----------------|-----------------|---------------|---------------|
> | HERN     | 10887.77        | 1313.08         | 2095.88       | 1051.85       |
> | MEAN     | 7162.65         | 2421.58         | 1041.43       | 1322.44       |
> | dyMEAN   | 3782.67         | 482.51          | 1730.06       | 544.91        |
> | DiffAb   | 1729.51         | 883.14          | 1297.25       | 1016.96       |
> | AbDPO    | 629.44          | 446.02          | 307.56        | 378.59        |
> | AbDPO+   | 1106.48         | 678.9           | 637.62        | 631.99        |
> | AbDPO++  | 1349.39         | 754.61          | 747.89        | 692.09        |
> | AbDPOw/O | 69.82           | 59.17           | -3            | 6.61          |

---

> > ### Comment · Reviewer_i9FY · 2024-08-12
> > **Thank you**
> >
> > Thank you for your response. I think the general methodology is applicable to properties other than rosetta. The new experiment shows the standard deviation and I encourage the authors to put them into the final paper. I have increased my score to 5.

---

> > > ### Author Response · Authors · 2024-08-13
> > >
> > > Thank you very much for your positive feedback. We will be sure to incorporate the deviation and other discussions into the final version. We greatly appreciate your recognition of our paper's soundness, presentation, and contributions, as reflected in your ratings for each of them. We would like to kindly remind you that a score of 5 is considered borderline and is advised to be used sparingly according to the review guidelines. We would really appreciate it if you could consider raising your score. Regardless of your final rating, we are sincerely grateful for your valuable suggestion and continued support.

---

### Author Rebuttal · Authors · 2024-08-07

We would like to thank all the reviewers for your constructive feedback. We placed Figure 1 and Figure 2 in the PDF file.

---

### Decision · Program_Chairs · 2024-09-25

**Decision:**

Accept (poster)

**Comment:**

After some discussion and extensive additional results provided by the authors, the reviewers' agree that the authors' method of combining energy based calculations and generative modelling is an interesting approach for optimizing binding energy. My only real comment here is to of course be sure to include the new results in the final version of the paper.